# Structural basis for differential inhibition of eukaryotic ribosomes by tigecycline

Xiang Li[1,3], Mengjiao Wang[1,3], Timo Denk [2,3], Robert Buschauer[2], Yi Li[1], Roland Beckmann [2] ✉ & Jingdong Cheng [1] ✉

Tigecycline is widely used for treating complicated bacterial infections for which there are no effective drugs. It inhibits bacterial protein translation by blocking the ribosomal A-site. However, even though it is also cytotoxic for human cells, the molecular mechanism of its inhibition remains unclear. Here, we present cryo-EM structures of tigecycline-bound human mitochondrial 55S, 39S, cytoplasmic 80S and yeast cytoplasmic 80S ribosomes. We find that at clinically relevant concentrations, tigecycline effectively targets human 55S mitoribosomes, potentially, by hindering A-site tRNA accommodation and by blocking the peptidyl transfer center. In contrast, tigecycline does not bind to human 80S ribosomes under physiological concentrations. However, at high tigecycline concentrations, in addition to blocking the A-site, both human and yeast 80S ribosomes bind tigecycline at another conserved binding site restricting the movement of the L1 stalk. In conclusion, the observed distinct binding properties of tigecycline may guide new pathways for drug design and therapy.

Tetracyclines are an important class of broad-spectrum antibiotics that inhibit bacterial translation by targeting their ribosomes[1,2]. Since the first tetracyclines, aureomycin, has been discovered and approved in the 1950s[3], tetracyclines have been widely used to treat human infections and even in animal feed[1,4]. Since then, more and more tetracycline resistance bacteria are emerging[5]. They either promote the efflux of the molecule (*tetA, tetB*, et al.), or develop ribosome protection proteins (*tetM*, et al.), or encode inactivating enzymes (*tetX*, et al.)[5–8]. In response to the growing rate of antibiotic resistance, new generations of tetracyclines were developed, among which is tigecycline[9]. It is the first FDA-approved drug (in 2005) that belongs to the third generation of tetracyclines (glycylcycline) and is structurally related to minocycline. Compared to minocycline, it has a *t*-butylglycylamido moiety at the 9th position on the D-ring. It has superior activity against many gram-positive, gram-negative and anaerobic bacteria, including methicillin-resistant *Staphylococcus aureus* (MRSA), *Stenotrophomonas maltophilia*, and multidrug-resistant strains of *Acinetobacter baumannii*[10]. Since then, it has been clinically prescribed for

the treatment of complicated skin and skin structure infections (cSSSI), complicated intra-abdominal infections (cIAI), community-acquired bacterial pneumonia (CABP), and as an emergency therapy for MDR pathogens[10].

Similar to tetracycline, tigecycline also inhibits bacterial growth by targeting their 70S ribosome[11–14]. It binds to the same site on h31 and h34 of the 16S rRNA that is the A-site tRNA binding site[11–13]. Thus, it sterically prevents the accommodation of the incoming A-site tRNA during the first step of translation[11–13]. However, due to the additional 9-*t*-butylglycylamido moiety, which forms a strong stacking with C1054 (*E. coli*) of the 16S rRNA, it binds to the 70S ribosome with ~20-fold higher affinity than tetracycline, even in the presence of the ribosome protecting protein TetM[11]. Consequently, tigecycline could overcome most tetracycline resistance mechanisms.

In addition to the known therapeutic uses as an antibiotic, tigecycline exhibits other non-antibiotic clinical activity, such as anti-inflammatory and the suppression of tumor metastasis activity[15].

[1]Minhang Hospital & Institutes of Biomedical Sciences, Shanghai Key Laboratory of Medical Epigenetics, International Co-laboratory of Medical Epigenetics and Metabolism, Fudan University, Shanghai, China. [2]Gene Center, Ludwig-Maximilians-Universität München, Munich, Germany. [3]These authors contributed equally: Xiang Li, Mengjiao Wang, Timo Denk. ✉e-mail: beckmann@genzentrum.lmu.de; cheng@fudan.edu.cn

Tigecycline has shown in vitro and in vivo activity against acute myeloid leukemia[16]. It also has anti-tumorigenic activity against several other types of cancer, including non-small cell lung cancer, gastric cancer, hepatocellular carcinoma, and glioblastoma[17]. Since the human mitoribosome is not only biochemically but also structurally similar to its bacterial ancestor, its anti-tumor activity is thought to be due to the inhibition of mitochondrial protein translation in eukaryotic cells. Notably, tumor cells with increased reliance on mitochondrial function exhibit increased sensitivity to tigecycline, making it an increasingly used agent in chemotherapy[16,17]. In addition, tigecycline exhibits more severe side effects and a growing number of resistant strains are isolated in the clinical setting[18]. Therefore, a comprehensive study is urgently needed to determine the mechanism of this inhibition, compare it with the inhibition of the bacterial 70S ribosome, and provide a molecular basis to guide future drug design efforts.

Here, we perform a comprehensive structural analysis of tigecycline binding to both the human 55S mitoribosome and the cytoplasmic 80S ribosome, and compare them with its interactions with yeast cytoplasmic 80S and bacterial 70S ribosomes[11–13]. We find that tigecycline may not only prevent the accommodation of the A-site tRNA, which is conserved with other members of the tetracycline family but also directly block the peptidyl transfer center (PTC) on the human 55S mitoribosome, which has not been observed before. In addition, we find that tigecycline also binds to the human 80S ribosome at high concentration at multiple sites, providing a structural basis for its only mild inhibitory activity on cytoplasmic translation.

## Results

### Tigecycline inhibits human translation by targeting the mitoribosome

We were encouraged to characterize the inhibitory properties of tigecycline on human ribosomes because of its increasing use in chemotherapy to kill human tumor cells as well as its high efficiency in inhibiting bacterial translation[1,7]. To monitor the translational activity of the ribosome in vivo, we used a quantification strategy based on the L-azidohomoalanine (L-AHA) labeling technique, which can be incorporated into proteins at the methionine position, for newly synthesized proteins[19,20]. To determine the inhibition of tigecycline on the 55S mitoribosome, cytosolic translation was selectively inhibited by pre-incubation with cycloheximide (50 μg/ml) followed by incubation with increasing concentrations of tigecycline. Compared to the CHX- and chloramphenicol-treated lane, the CHX-only treatment still shows mitoribosome-translated protein bands, while with increasing tigecycline, the translation activity of the 55S mitoribosome is potently inhibited with an IC$_{50}$ (half maximal inhibitory concentration) of approx. 0.6 μM, which is comparable to that on the bacterial 70S ribosome[11] (Fig. 1a−c). This finding indicates that tigecycline effectively targets the human mitoribosome.

In contrast, in the case of cytoplasmic translation, our results show that tigecycline only mildly inhibited the translational activity of the cytoplasmic 80S ribosome (Fig. 1b, c). Unfortunately, we cannot measure its IC$_{50}$ since we could not completely inhibit it, indicating that the human cytoplasmic 80S ribosome is relatively resistant to tigecycline, similar to its described resistance to tetracycline[21]. To exclude the effects which could be introduced by tigecycline on other

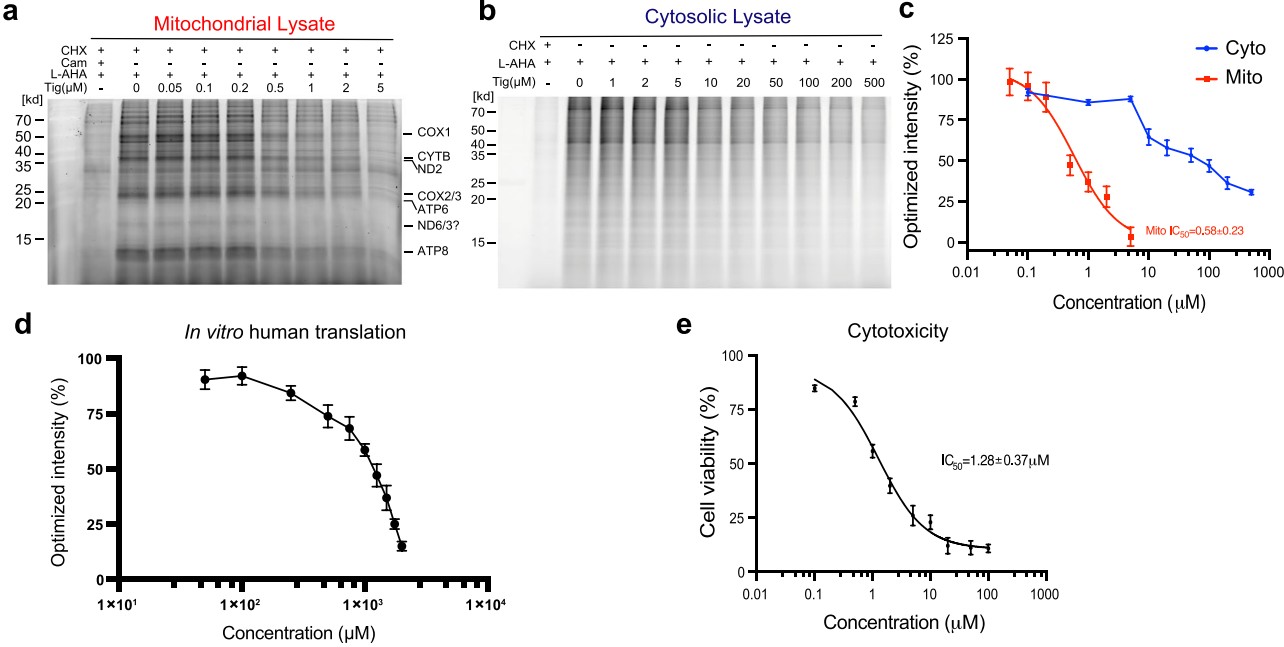

**Fig. 1 | Inhibitory properties of tigecycline on human ribosomes. a−c** De novo mitochondrial (**a**, **c**) and cytoplasmic (**b**, **c**) nascent protein synthesis were measured by L-AHA labeling, followed by visualization of Alexa Fluor 488 fluorescence signal conjugated to L-AHA by "click reaction". The intensity of the corresponding lines in the SDS-PAGE was analyzed using the ImageJ software. To estimate the IC$_{50}$ value of tigecycline on the 55S mitoribosome in the mitochondrial lysate (Mito) and 80S human ribosome in the cytosolic lysate (Cyto), the intensities of positive controls derived from chloramphenicol (Cam) and CHX-treated samples, respectively, were subtracted. Second, the resulting intensities were normalized to the intensities of the negative controls, which in both cases were from the samples not treated with tigecycline. Note that tigecycline cannot sufficiently inhibit cytoplasmic translation even at high concentrations. Error bars (standard deviation) were calculated from at least three groups of protein bands whose intensities were quantified using ImageJ software (for mito, n = 4 groups; for cyto, n = 3 groups), and data were presented as mean values +/− SD. **d** In vitro translation of Nluc mRNA in HEK293T whole cell lysate in the presence of increasing concentrations of tigecycline. Error bars (standard deviation) were calculated from three different experimental units (n = 3 independent experiments), and data were presented as mean values +/− SD. **e** Cell viability of HEK293T cells in the presence of increasing concentrations of tigecycline, as measured by CCK-8 assay. Error bars (standard deviation) were calculated from three different experimental units (n = 3 independent experiments), and data were presented as mean values +/− SD. Source data are provided as a Source Data file.

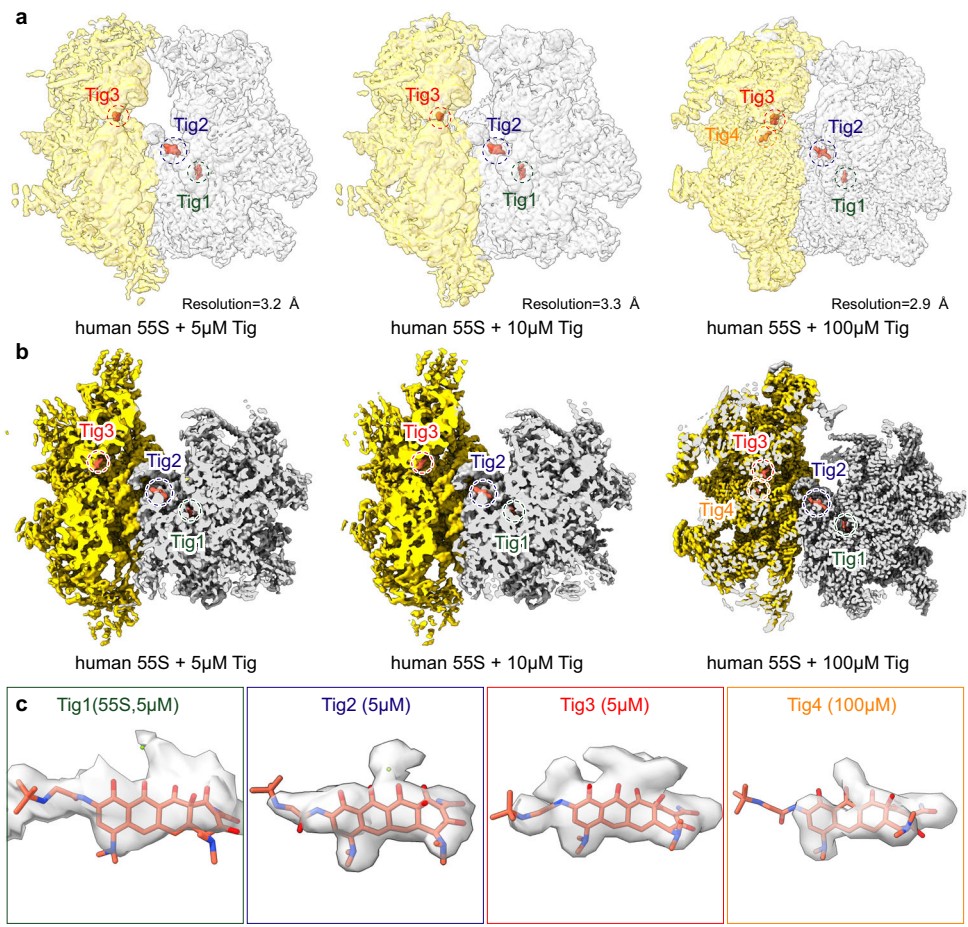

**Fig. 2 | Cryo-EM structures of tigecycline on the human mitoribosome.**
**a**, **b** Overview (**a**) and cross-section (**b**) of the cryo-EM maps of the 55S mitoribosome (28S, yellow; 39S, gray) showing the binding sites of four tigecycline molecules (at various concentration levels, tomato red) within both the large and small subunits. The overview (**a**) of the 55S mitoribosome densities are shown in translucent, while the densities of the tigecycline molecules are not. **c** The details of tigecycline densities are shown. The Tig1-3 binding sites were consistent across all concentrations, except for Tig4, which only appeared under the 100 μM tigecycline condition. Densities for Tig2-4 are derived from the cryo-EM density map postprocessed by DeepEMhancer, whereas the density of Tig1 were derived from the consensus map without post-processing.

cell activity than translation, we used an in vitro human cell translation assay and monitored its inhibition in vitro[22]. Consistently, this in vitro result is in agreement with our in vivo findings (Fig. 1d) and implies that while tigecycline primarily targets the human 55S mitoribosome, it also disturbs the activity of the 80S ribosome to some extent, albeit with low binding affinity and thus only at higher concentrations.

To characterize the overall cytotoxicity of tigecycline, we treated the human cells with increasing amounts of tigecycline and monitored the viability of the cells. In our context, the IC$_{50}$ was about 1.3 μM (Fig. 1e), which is similar to the IC$_{50}$ for mitochondrial translation inhibition and corresponds to the physiological concentration during clinical treatment (the mean serum tigecycline concentration can reach about 4.8 μM after a single dose)[23–25]. In comparison with the individual inhibitory properties on both the human 55S mitoribosome and the cytoplasmic 80S ribosome, we conclude that tigecycline primarily targets the 55S mitoribosome during clinical administration as suggested before[16].

**Cryo-EM structures of tigecycline with human mitoribosomes**
Inspired by the strong inhibition of tigecycline on the human mitoribosome, we solved six structures of human 55S and 39S mitoribosomes incubated with different concentrations (5 μM, 10 μM, and 100 μM) of tigecycline by single-particle cryo-EM analysis (Fig. 2a, b and Supplementary Figs. 1–5). When the human mitoribosomes were

exposed to 100 μM tigecycline, the resulting cryo-EM densities facilitated us to assign four tigecycline molecules (Tig1-4) to the 55S mitoribosome (2.9 Å resolution, Supplementary Figs. 1–5 and Supplementary Table 1). In contrast, when the human mitoribosomes were exposed to 5 μM or 10 μM tigecycline, concentrations within the physiological range during clinical usage[23–25], only three tigecycline molecules (Tig1-3) could be distinctly assigned, with Tig4 absent (Fig. 2a, b and Supplementary Fig. 5a). This discrepancy suggests that the binding of Tig4 occurs specifically under high concentrations of tigecycline.

All tigecycline molecules on the 55S ribosome are well resolved, including their characteristic coordinated magnesium ion (Fig. 2c and Supplementary Fig. 5a). Two of the tigecycline molecules (Tig3 and Tig4) associate with the 28S small subunit located within the mRNA-binding tunnel near the A-site tRNA binding region (Fig. 2a, b and Supplementary Fig. 5a). Meanwhile, the remaining two tigecycline molecules associate with the 39S large subunit, with one (Tig1) binding directly to the peptidyl transfer center (PTC) and the other one (Tig2) binding in the vicinity (Fig. 2a, b and Supplementary Fig. 5a). Unfortunately, the resolution of the tigecycline molecule (Tig1) on the 39S mitoribosome was not optimal. However, based on the assignment of the corresponding Tig1 molecule on the 55S mitoribosome, we attempted to identify this additional density in the same region as a tigecycline molecule in the 39S mitoribosome (Supplementary Fig. 5b).

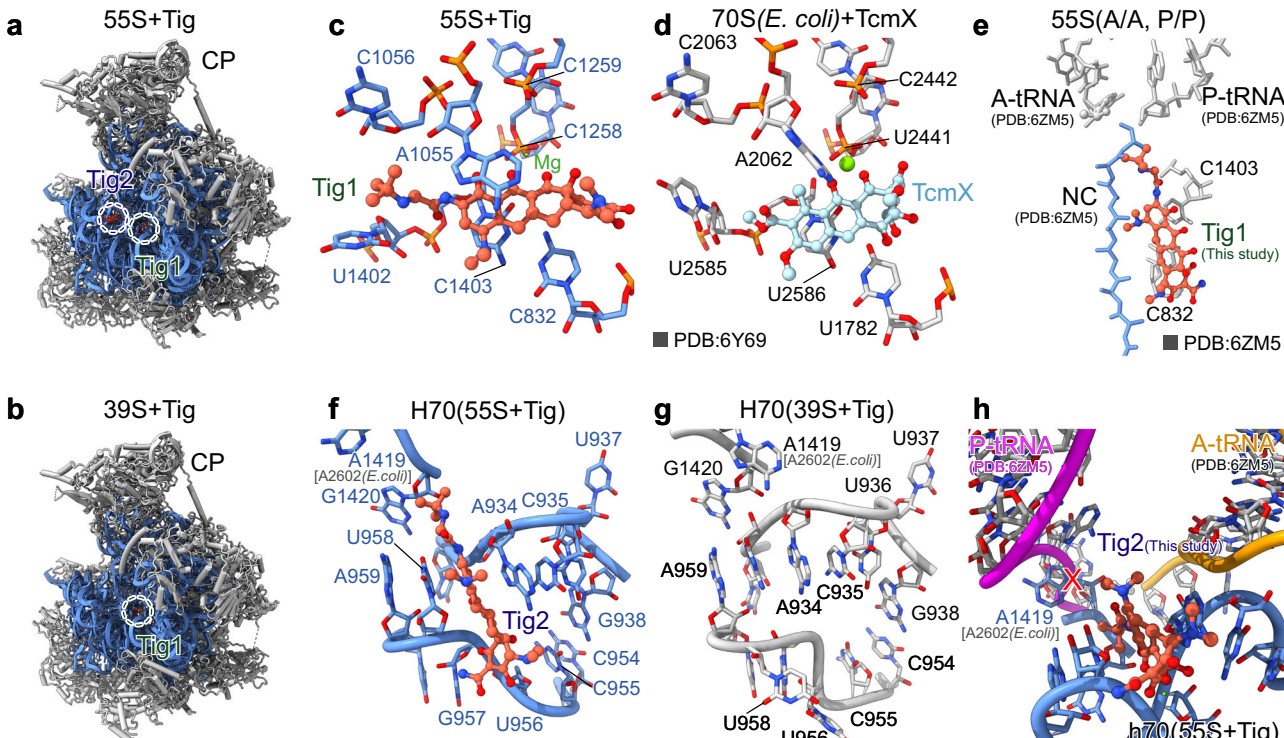

**Fig. 3 | Tigecycline blocks PTC on the human mitoribosome. a, b** Overview of the 39S large subunit models of 55S (**a**) and 39S (**b**) mitoribosomes (16S rRNA, blue; 39S mitoribosomal proteins, gray) showing the Tig1 and Tig2 molecules (tomato red) within the peptidyl transfer center (PTC). The model originates from the human mitoribosome treated with 100 μM tigecycline. **c** The Tig1 molecule interacts in the PTC with the presence of a putative Mg ion (green). **d** Comparison with TcmX interaction in the PTC of the *E. coli* 70S ribosome. **e** The 55S mitoribosome with tigecycline structure compared to the translating one with A- and P-site tRNA and nascent peptide chain, indicating the potential clash of the Tig1 molecule with the nascent peptide in the peptide exit tunnel (PDB 6ZM5). **f** The detailed structure of the Tig2 molecule binding to H70 region of the 16S rRNA in the 55S mitoribosome with tigecycline. **g** The corresponding H70 region of the 16S rRNA in the 39S mitoribosome with tigecycline, indicating the absence of the Tig2 molecule. **h** The Tig2 molecule could affect the translocation of the tRNA from the A-site to the P-site. The conformation of base A1419 (A2602 in *E. coli*) would clash with the accommodated P-site tRNA. All key nucleotide bases are labeled. In some cases, the corresponding nucleotide bases in the *E. coli* 70S ribosome are also shown in gray.

## Tig1 and Tig2 bind to the mitoribosomal PTC

The mechanism of inhibition of tigecycline on the bacterial 70S ribosome, similar to that of tetracycline, involves tight binding to the 30S small subunit, thereby preventing the accommodation of incoming A-site tRNA[11–14]. Unexpectedly, however, two of the tigecycline molecules, Tig1 and Tig2, bind to the 39S large subunit in the 55S mitoribosome (Fig. 3a–h).

The Tig1 molecule binds directly to the PTC on the 39S large subunit (Fig. 3a). It is sandwiched between base A1055 and non-canonical C1403:C832 base pair (Fig. 3c). Specifically, the D-ring forms stable π-π stacking with both nucleobases A1055 and C1403, while a magnesium ion bridges the B-C-ring interaction with the C1258-C1259 backbone (Fig. 3c). The 9-*t*-butylglycylamido moiety of tigecycline points directly into the PTC, with the *t*-butyl group binding in close proximity to the nucleobases A1402 (*E. coli* A2585) and C1056 (Fig. 3c), and would directly clash with the nascent chain (Fig. 3e). Thus, tigecycline could directly impair the PTC since A1402 is responsible for hydrolyzing the tRNA ester bond[26,27]. When carefully compared with the available antibiotic ribosome structures, we find that Tig1 binds to the same site as its structurally related tetracenomycin X (TcmX) on the *E. coli* 70S ribosome or the human 80S ribosome[22,28] (Fig. 3d), which indicates that tigecycline might also has potential to block protein synthesis in a nascent chain depended manner on human 55S mitoribosomes[28]. While TcmX exclusively binds to ribosomes which have the U:U non-canonical base pair (Fig. 3c, d)[22,28], tigecycline solely interacts with the 30S small subunit on these ribosomes[11–13]. However, on the human 55S mitoribosome, which has a C–C base pair,

tigecycline binds not only to the 28S small subunit but also to the corresponding position. Thus, it is reasonable to speculate that the non-canonical base pair at this position determines the specificity of the antibiotic binding to it.

The Tig2 binds to H70 of the 16S rRNA, yet only on the 55S mitoribosome (Fig. 3b, f, g). Since this region is right next to the inter-subunit bridges B2 and B3[29], the joining of the 28S small subunit likely influences this region in order to accommodate the tigecycline. When comparing the 16S rRNA Helix 70 region of the 55S and 39S mitoribosome, the binding of the Tig2 molecule induces a large conformational change, including the rearrangement of the bases U956–U958, C933–A934 and A1419 (*E. coli* A2602) (Fig. 3f, g). As a result, tigecycline stacks in between bases U958 and A934, with its 9-*t*-butylglycylamido moiety pointing directly toward A1419, forcing it into an outward position (Fig. 3f, g). Although Tig2 does not directly bind to the PTC, it is located directly in the path of the tRNA as it translocates from the A-site to the P-site (Fig. 3h). Moreover, A1419 (*E. coli* A2602), well-known for its conformational changes and role in peptide formation[26,30,31], in this outward conformation could collide with the accommodated P-site tRNA (Fig. 3h). Taken together, Tig1 and Tig2 are likely to impair the mitoribosomal PTC and, in addition, to prevent tRNA translocation.

## Tig3 and Tig4 bind to the A-site on 28S small subunit

In addition to binding to the 39S large subunit, similar to tigecycline on the bacterial 30S ribosome[11–13], tigecycline also binds to the 28S small subunit (Fig. 4a–e). In total, we identified two tigecycline molecules,

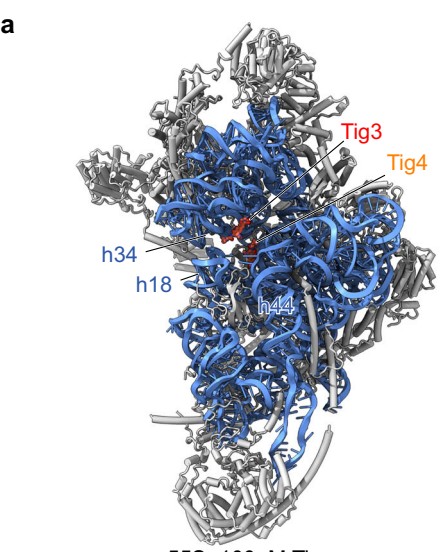

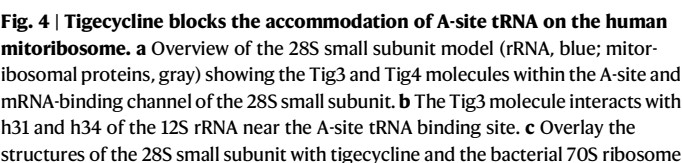

**Fig. 4 | Tigecycline blocks the accommodation of A-site tRNA on the human mitoribosome. a** Overview of the 28S small subunit model (rRNA, blue; mitoribosomal proteins, gray) showing the Tig3 and Tig4 molecules within the A-site and mRNA-binding channel of the 28S small subunit. **b** The Tig3 molecule interacts with h31 and h34 of the 12S rRNA near the A-site tRNA binding site. **c** Overlay the structures of the 28S small subunit with tigecycline and the bacterial 70S ribosome with tigecycline (PDB:4V9B) shows that the Tig3 molecule binds to the same "primary site" as the one (cyan) in the bacterial 70S ribosome. **d** The Tig4 molecule interacts with the h1 and h44 regions of the 12S rRNA within the mRNA-binding channel. **e** Superimposition of the 28S small subunit with tigecycline structure with the translation one (PDB 6ZM5) shows that the Tig3 and Tig4 molecules block the recognition of the A-tRNA anticodon and the mRNA codon, respectively.

Tig3 (the primary binding site) and Tig4 (only present in the 100 μM tigecycline incubated sample, Fig. 4a) bind to the 28S small subunit.

Since the universal conserved "primary" binding site for tetracycline and its derivatives[11–14] on the bacterial 70S ribosome is identical to that of the 28S small subunit[11–13], we find the Tig3 molecule bound to h34 of the 12S rRNA as expected (Fig. 4b, c). When compared to tigecycline on the bacterial 70S ribosome, this molecule adopts almost the same conformation, with the only small difference in the 9-*t*-butyl-glycylamido moiety (Fig. 4b, c).

Tig4 binds to a region near h1 and h44 of the 12S rRNA (Fig. 4d), exclusively observed in the 55S mitoribosome treated with 100 μM tigecycline. We speculate this is a potential binding site for tigecycline under certain conditions, since it was found as a second tetracycline binding site for negamycin-resistant strain carrying U1052G mutation[32]. Because of this "secondary" binding site, this resistant strain shows hyper susceptibility towards tetracycline, highlighting the importance of this binding site[32]. Here, the D-ring and the 9-*t*-butyl-glycylamido moiety are tightly stacked on nucleobase C834 (*E. coli* C1397), while the A-B-rings interact with the nucleobases from h1 of the 12S rRNA and R237 of the MRPS5 on the 55S mitoribosome (Fig. 4d). At this position, Tig4 would collide with the codon of the mRNA during decoding by the A-site tRNA (Fig. 4e). Meanwhile, the classical "primary" binding site (Tig3) would collide with the anticodon on the A-site tRNA (Fig. 4e). Taken together, bound at both sites, Tig3 and Tig4 could sterically prevent the accommodation of the incoming A-site tRNA employing a redundant mechanism.

## Tigecycline does not bind to human 80S ribosomes at physiological conditions

Although tigecycline inhibits human translation by targeting mitoribosomes, the mild inhibition of tigecycline on the human 80S ribosome has remained puzzling. To address this point, we systematically investigated the interaction between tigecycline at different concentrations and empty (without translational factors) human 80S ribosomes. In detail, human 80S ribosomes were disassembled into subunits by high-salt and puromycin treatment, and then reassociated

to form factor-free 80S ribosomes. These purified empty 80S ribosomes were then incubated with two concentrations of tigecycline, including clinically relevant concentrations (4 μM) and higher concentrations (100 μM).

Consistent with our biochemical data, our cryo-EM single-particle analysis reveals that tigecycline does not bind to human 80S ribosomes at 4 μM tigecycline (Figs. 1b–d, 5a, Supplementary Fig. 6, and Supplementary Table 2), indicating that tigecycline does not inhibit human 80S ribosomes during clinical usage. However, at higher concentrations, we observed five tigecycline molecules binding to the 80S ribosome (Fig. 5b). Among these, Tig4 and Tig5 were found to bind to the peripheral region of the 60S ribosome (Fig. 5b), unlikely to affect translation. In contrast, Tig3 was located in the "primary" binding site as observed on bacterial 70S ribosomes and human 55S mitoribosomes (Figs. 3c, d and 5c), indicating a conserved mode of binding across prokaryotic and eukaryotic ribosomes.

Furthermore, Tig1 and Tig2 were observed to form a dimer near the L1 stalk region (including H76 of the 28S rRNA) of the 80S ribosome (Fig. 5d). These two tigecycline molecules insert into a cleft between H68 and H76 of the 28S rRNA, bridging the two helices and potentially restricting L1 stalk movement (Fig. 5d, e). The L1 stalk region is known for its dynamic motion during the translation elongation cycle, facilitating tRNA and mRNA translocation through the ribosomal complex. The binding of Tig1 and Tig2 to this region suggests a role in modulating ribosomal conformational dynamics during translation. Indeed, during translocation, as the 40S body rotates, h23 of the 18S rRNA passes through this site, with nucleobase G970 (human 18S rRNA) even pointing into this site (Fig. 5e). Therefore, the association of Tig1-3 molecules only at high concentrations may explain its mild inhibiting effect on cytoplasmic translation.

## Tigecycline binds to multiple sites on the native eukaryotic 80S ribosomes

To further explore the effect of tigecycline on cytoplasmic translation, we purified native 80S ribosome complexes from human HEK293 cells treated with a high dose of tigecycline (170 μM). Using cryo-EM

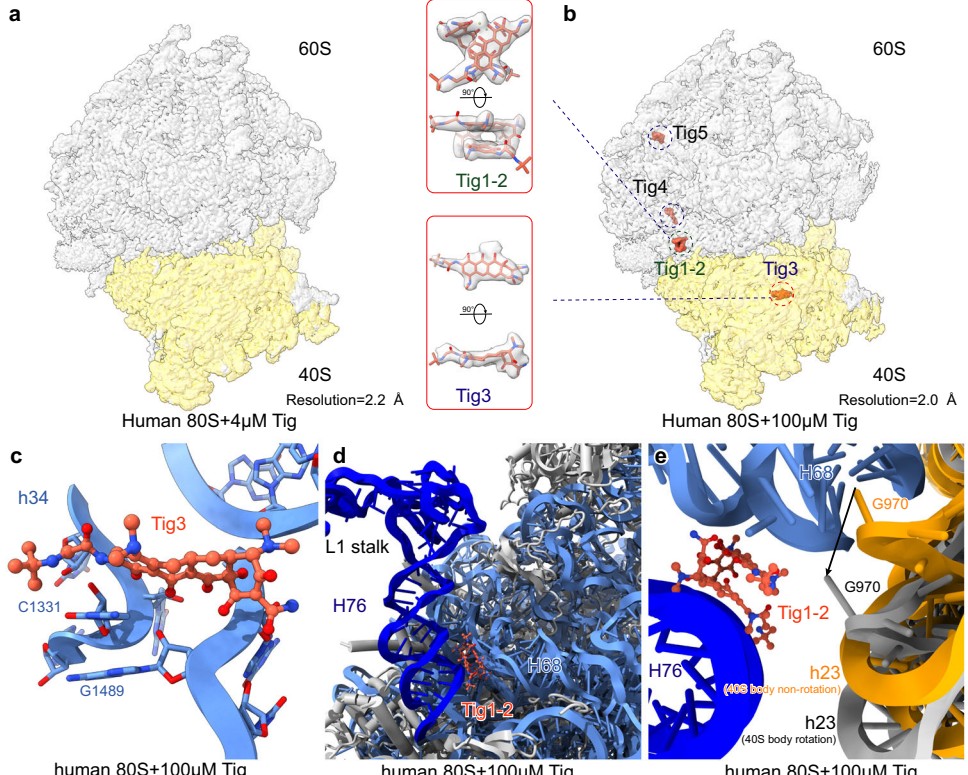

**Fig. 5 | Cryo-EM structure of tigecycline on the human 80S ribosome.**
**a**, **b** Overview of the cryo-EM structures of the human 80S ribosome incubated with either 4 μM (**a**) or 100 μM (**b**) tigecycline. The density of the 80S ribosome is shown as translucent, while the density of the tigecycline molecules is not. Two different views of Tig1-3 molecules are also shown as middle inserts. **c** The "primary" binding site (h31 and h34 region of 18S rRNA, blue) in human 80S ribosome. The tig3 molecule stacks well with base C1331 of the 28S rRNA. **d** Detailed structures of the Tig1-2 (tomato red) molecules binding sites within human 80S ribosomes near the L1 stalk. **e** The Tig1 and Tig2 molecules within the human 80S ribosome may affect the movement of 18S rRNA helix 23, thereby influencing 40S body rotation.

single-particle analysis and extensive 3D sorting, we obtained four different ribosome structures with different numbers of bound tige-cycline molecules (Fig. 6, Supplementary Figs. 7 and 8, and Supple-mentary Tables 2 and 3): 1) the human 80S+Tig (with E-tRNA and CCDC124) state; 2) the human 80S+Tig (with E-tRNA and CCDC124, 40S head swiveled) state; 3) the human 80S+Tig (E-tRNA, eEF2, and SERBP1) state; and 4) the human 80S+Tig (with E-tRNA, P-tRNA, and mRNA) state. These structures represent different translational states, characterized by different rotations of the 40S ribosome body or head, accompanied by different translation factors (Fig. 6a).

In the human 80S+Tig (with E-tRNA and CCDC124) state, we identified six tigecycline binding sites housing a total of nine tigecy-cline molecules (Tig1-10, except Tig3) on the human 80S ribosome (Fig. 6b). Notably, most of these molecules were located in the non-conserved peripheral region, where the translation is less likely to be affected (Fig. 6b, Supplementary Fig. 9, and Supplementary Table 3). Tig10, binding to the E-tRNA and the 40S head, was exclusive to this state and absent in others (Fig. 6b, Supplementary Fig. 9d, e, and Supplementary Table 3). Conversely, Tig9, also associated with E-tRNA in the both human 80S+Tig (with E-tRNA and CCDC124) and human 80S+Tig (E-tRNA, eEF2, and SERBP1) states (Fig. 6b, Supplementary Fig. 9b, c, and Supplementary Table 3). Speculatively, Tig9-10 repre-sents E-site tRNA and 40S head conformation-dependent binding sites.

In the human 80S+Tig (with E-tRNA, P-tRNA, and mRNA) state, representing an actively translating 80S ribosome, only the tigecycline molecules (Tig4-8) bound to the peripheral region persisted (Fig. 6b, Supplementary Fig. 9f–h, and Supplementary Table 3), suggesting that these sites do not affect translation. The absence of Tig1-3 mole-cules in this actively translating ribosome implies that Tig1-3 may exert

only a mild influence on human translation (Supplementary Table 3), consistent with our analysis of tigecycline with empty 80S ribo-somes (Fig. 5).

To validate these findings on the human 80S ribosome, we also solved the cryo-EM structure of the tigecycline complex with the yeast 80S ribosome from one of our published datasets[33] (Fig. 6, Supple-mentary Fig. 10, and Supplementary Tables 2 and 3), where a similar high-dose treatment (170 μM) was used. Unlike the reported struc-tures, we selected the dataset derived from the tigecycline-stalled monosome sample[33]. We obtained two different high-resolution yeast 80S ribosome structures, one state with eEF2/Stm1/eIF5A binding and the other state with Not5 and P-site tRNA binding.

We found four tigecycline binding sites housing a total of six tigecycline molecules on the yeast 80S ribosome (Fig. 6, Supplemen-tary Fig. 11, and Supplementary Tables 2 and 3). Interestingly, these six tigecycline molecules (Tig1-3, 11-13) also form dimers and trimers, and are mostly located at the peripheral region of the yeast 80S ribosome (Fig. 6 and Supplementary Fig. 11). Consistent with a previous bio-chemical study[34] and our analysis of the human 80S ribosome (Fig. 5), we find a tigecycline molecule (Tig3) binding in the A-site (Supple-mentary Fig. 11c) in the yeast 80S+Tig (with Not5, P-tRNA and mRNA) state, which explains its canonical tRNA accommodation inhibition activity[34]. Furthermore, we find that in both human and yeast 80S ribosomes, the associated tRNA is the initial methionine tRNA (Sup-plementary Fig. 12), which could be a result from the Tig3 molecule blocking the incoming A-site tRNA after initiation[14].

Moreover, the additional binding site of Tig1-2 observed in the yeast 80S+Tig (with Not5, P-tRNA and mRNA) state or Tig1-3 in the yeast 80S+Tig (with eEF2, Stm1, eIF5a) state is conserved compared to

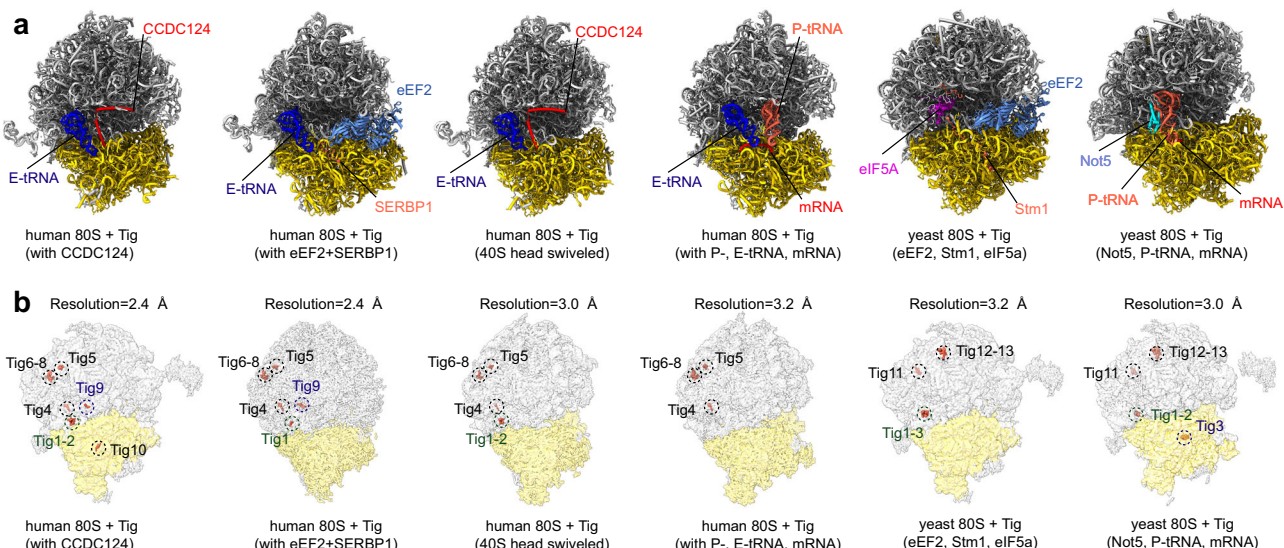

**Fig. 6 | Cryo-EM structure of tigecycline on the eukaryotic 80S ribosome. a** The sliced view of the human and yeast 80S ribosome structures. Translation factors are highlighted in distinct colors. **b** Overview of cryo-EM maps of human and yeast 80S ribosomes (40S, yellow; 60S, gray) showing nine and six tigecycline (tomato red) molecules within the large and small subunits, respectively. The 80S ribosome densities are shown in translucent, while the densities of the tigecycline molecules are not. All densities are derived from the cryo-EM density map post-processed by DeepEMhancer.

the binding site of Tig1-2 on the human ribosome, which is located near the L1 stalk and the E-site tRNA (Fig. 5d and Supplementary Fig. 11d). This suggests that high concentration of tigecycline might inhibit eukaryotic 80S ribosomes through two conserved binding sites: one is the canonical "primary" binding site of Tig3, and the other is the L1 stalk region of the Tig1-2 molecules.

## Discussion

In this study, we found that tigecycline inhibits human translation primarily by targeting mitochondrial ribosomes. We observed multiple tigecycline binding sites on human 55S mitoribosomes. Among them, one site on the 28S small subunit is conserved with the site where tigecycline binds to the bacterial 70S ribosome, but all the other three sites on the 55S mitoribosome have not been observed so far. Multiple binding sites have been observed for other antibiotics, e.g., kasugamycin, with two binding sites all located inside the mRNA-binding tunnel[35], and neomycin, altering the rotation of the small subunit with one binding to H44 of the 16S rRNA and the other to H69 of the 23S rRNA[36]. This is very likely resembling the binding mode of our Tig2 and Tig3 on the 55S mitoribosome.

Undoubtedly, the Tig1 binding site on the 55S/39S mitoribosome is an interesting site. It has never been observed before, including on the bacterial 70S ribosome and the eukaryotic 80S ribosomes. A previous extensive biochemical study on TcmX has already demonstrated that the noncanonical U−U base pair in the PTC determines the binding property of TcmX[22]. It is also plausible to speculate that the noncanonical C−C base pair may be able to determine the binding property of tigecycline. Based on available rRNA data, in the bacterial world, 63% of species have a corresponding C−C base pair at this position, whereas only 37% have a U−U base pair required for TcmX binding[22]. This could explain the much broader antibacterial spectrum of tigecycline compared to TcmX. Therefore, a combination treatment with both tigecycline and tetracenomycin X may serve as a new therapy with a much broader effect.

While the Tig3 molecule binds to the conserved "primary" binding site as in the bacterial 70S ribosome, the Tig2 molecule is exclusively observed binding to H70 of the 16S rRNA in the human 55S mitoribosome. This distinction may arise from the lack of conservation of H70 between the 55S mitoribosome and the bacterial 70S ribosome, particularly the significant A934 stacking with the tigecycline rings (C1941 in *E. coli*). The Tig4 molecule on the other hand is only detected in the human 55S mitoribosome treated with 100 μM tigecycline, suggesting that Tig4 binds only under high concentrations of tigecycline.

Furthermore, in human 80S ribosomes, two other binding sites, Tig9 and Tig10, interact either directly with E-site tRNA or very close to the anticodon stem, which could also contribute to inhibition (Supplementary Fig. 9b–d) at certain translation stages. Contrary, we do not observe E-site tRNA in our yeast 80S ribosome. Therefore, we cannot confirm the conservation of these two sites. These observations raise the possibility that tigecycline could inhibit the ribosome during specific translational states or in dependence on particular translation factors, characteristics that may be shared with other antibiotics. Therefore, this could herald a novel strategy for antibiotic design.

It is interesting to note that tetracycline itself has been observed to have multiple binding sites on the 30S small subunit[32,37]. Similar to our structure of tigecycline with the 80S ribosome, most of the "secondary" binding sites are not located at translation-affecting positions. The existence of these "secondary" binding sites most possibly being a result of incubation with high concentrations of tigecycline (100 μM), however, our observations reveal conserved binding sites between human and yeast. Despite the non-physiological concentration of tigecycline used, studying the inhibition mechanism at high concentrations remains valuable for research purposes[34] or to guide new drug design.

It is important to note that our structural analysis provides predictions of the inhibition mechanism, but further functional studies are necessary to validate these hypotheses.

## Methods

### Cell lines and cell culture

HEK293T or SK-Hep1 cells were cultured at 37 °C in 5% $CO_2$ in DMEM (BasalMedia, L110KJ) supplemented with 10% FBS (Yeasen Bio, 40130ES76) and 100 U/mL penicillin/streptomycin (BasalMedia, S110JV). Expi293 cells were cultured in SMM 293-TI expression medium

(Sino Bio, M293TII) supplemented with 100 U/mL penicillin/streptomycin. L-methionine-depleted DMEM (-Met) medium was purchased from Gibco (Thermo Fisher, 21013024).

## L-AHA labeling on nascent mitochondrial-encoded proteins

HEK293T cells were seeded in 10 cm dishes until cell confluence reached ~80%. Prior to labeling, the cells were washed once with DMEM (-Met) medium and then incubated in fresh DMEM (-Met) medium for another 30–60 min at 37 °C in 5% $CO_2$. The cells were treated with different concentrations of tigecycline (MCE, HY-B0117) to inhibit mitochondrial translation. Meanwhile, cytosolic translation was stopped by treating the cells with cycloheximide (CHX, 50 μg/ml, MCE, HY-12320). In addition, a plate of cells treated with 150 μg/ml chloramphenicol (Cam, Meilunbio, MB2014) was used as a positive control. After 30 min of incubation, 50 μM L-azidohomoalanine (L-AHA, MCE, HY-140346A) was added and the cells were incubated for another 3–4 h. After labeling, cells were gently scraped with a cell scraper and washed once with cold PBS buffer supplemented with 1 mM EDTA, followed by isolation of mitochondria by differential centrifugation. Briefly, cells were resuspended in mitochondrial isolation buffer (MTiso buffer) containing 3 mM HEPES, 210 mM mannitol, 70 mM sucrose, 0.2 mM EGTA and protease inhibitor (Roche, 4693132001). Cells were then homogenized on ice using a Dounce homogenizer (1 ML, DURAN WHEATON KIMBLE) and the homogenate was piled up on an equal volume of 340 mM sucrose in new tubes. After two centrifugation steps, one at $500 \times g$ for 10 min at 4 °C and another at $1000 \times g$ for 10 min at 4 °C, to remove cell debris, mitochondria were collected by centrifugation at $10,000 \times g$ for 10 min at 4 °C. Freshly isolated mitochondria were washed once with MTiso buffer and then lysed with click lysis buffer containing 50 mM Tris-HCl (pH 8.0) and 1% SDS for 30–60 min on a rotator at 4 °C. The click reaction was performed using a commercial kit (Click-iT Cell Reaction Buffer Kit; Thermo Fisher, C10269) with 5 μM Alexa Fluor 488-alkyne (Sigma, 761621) for ~1–2 h at room temperature away from light. The reaction was stopped by the addition of SDS-PAGE loading buffer. Final proteins were separated on 16% SDS-PAGE gels. Fluorescence signals in the gels were analyzed using an Amersham Typhoon 5 (Cytiva). All experiments were performed in triplicate ($n = 3$ independent experiments).

## L-AHA labeling on nascent cytosolic proteins

Nascent cytosolic protein labeling was done the same as on nascent mitochondrial-encoded proteins. Only the following is different: cells treated with cycloheximide (CHX, 50 μg/ml, MCE, HY-12320) were used as a positive control. After labeling, cells were harvested and resuspended with LSB-buffer containing 60 mM Tris-HCl (pH 8.0), 50 mM NaCl, 2 mM $MgCl_2$, 40 mM KCl, 5% glycerol, 0.1% NP-40 and protease inhibitor (Roche, 4693132001). Cells were subsequently homogenized on ice using a dounce homogenizer (1 ML, DURAN WHEATON KIMBLE). The cell lysate was cleared by centrifugation at $15,000 \times g$ for 15 min at 4 °C, and the supernatant was transferred to new 1.5 mL tubes. The click reaction and the following protein signal detection were done the same as above. All experiments were performed in triplicate ($n = 3$ independent experiments).

## In vitro transcription

The template for in vitro transcription was PCR amplified from plasmid pT7CFE1-NLuc. The capped NLuc reporter mRNA was generated using the mMESSAGE mMACHINE T7 Transcription Kit (Invitrogen, AM1344) according to the manufacturer's instructions. After transcription, the remaining template DNA was removed by DNase treatment, and the mRNA was purified by LiCl precipitation.

## In vitro translation assay

In vitro translation extract was prepared similarly as described before with certain changes[38]. Suspension-adapted HEK293T cells were grown to a density of $2.0 \times 10^6$ cells/mL in HyCell TransFx medium (Cytiva, SH30939) supplemented with 0.01% Poloxamer 188 (Sigma Aldrich, P5556), 1× penicillin–streptomycin (Gibco, 15140122) and 3× GlutaMAX (Gibco, 35050061). Cells were treated with 200 nM ISRIB for 1 h before harvesting to ensure sufficient cap-dependent translation[39]. Then, cells were pelleted and subsequently washed with Washing Buffer (35 mM HEPES-KOH pH 7.5, 140 mM NaCl, 20% w/v glucose) and Extraction Buffer (20 mM HEPES-KOH pH 7.5, 45 mM KOAc, 45 mM KCl, 1.8 mM $MgCl_2$, 1 mM DTT). The resulting pellet was frozen in liquid nitrogen as droplets and subjected to cryogenic grinding with a 6970EFM Freezer/Mill (SPEX SamplePrep). The cell powder was resuspended and thawed in 1 mL Extraction Buffer per $1.2 \times 10^9$ cells. High Potassium Buffer (20 mM HEPES pH 7.5, 945 mM KOAc, 945 mM KCl, 1.8 mM $Mg(OAc)_2$, 1 mM DTT) equal to 1/29 of total lysate volume was added and the lysate was incubated for 5 min on ice. The extract was consecutively cleared by centrifugation in a TLA-110 rotor (Beckman Coulter) at $15,700 \times g$ for 13 min and at $74,300 \times g$ for 19 min. To remove endogenous mRNAs the extract was further treated with 75 U/mL S7 nuclease and 1 mM $CaCl_2$ at 23 °C for 5 min. The reaction was stopped by the addition of 2.4 mM EGTA. Aliquots of the in vitro translation extract were frozen in liquid nitrogen and stored at −80 °C.

For the in vitro translation assay 10 μL reactions with 50% v/v cell extract were adjusted to 20 mM HEPES pH 7.5, 0.42 mM $MgCl_2$, 1.25 mM $Mg(OAc)_2$, 75 mM KOAc, 37.5 mM KCl, 1 mM DTT, 1.56 mM GTP, 0.25 mM ATP, 1.6 mM creatine phosphate, 0.45 mg/mL creatine kinase, 50 μg/mL bovine liver tRNA (Sigma Aldrich), 0.4 mM spermidine, 0.12 mM amino acid mix (homemade) and 0.8 U/μL SUPERase-In RNase inhibitor (Invitrogen, AM2694). Reaction mixes were incubated with different concentrations of tigecycline for 20 min on ice. The translation reaction was initiated by the addition of 250 ng NLuc reporter mRNA and carried out at 30 °C for 2 h. Subsequently, the reactions were diluted in 200 μL nuclease-free $H_2O$. In total, 50 μL diluted reaction were mixed with 50 μL NLuc substrate in NanoGlo Buffer (Promega, N1110) in a 96-well plate. The resulting luminescence signal was measured on an Infinite M1000 (Tecan).

## CCK-8 cell growth assay

Cytotoxicity was measured in both HEK293T and SK-Hep1 cell lines using the CCK-8 cell growth assay. Initially, 2000 cells per well (HEK293T) or 500 cells per well (SK-HEP1) were seeded in a 96-well plate in DMEM medium supplemented with 10% FBS and 1% penicillin/streptomycin. After 24 h of growth, the cells were treated with different concentrations of tigecycline or DMSO. After 72 h of incubation, the original medium was discarded and replaced with fresh medium supplemented with CCK-8 (added at a ratio of 100:10). The cells were then incubated for an additional 2 h in the dark. Finally, the cell growth rate was calculated by measuring the absorbance of each well at 450 nm. The cytotoxic concentration that inhibits 50% viability ($IC_{50}$) of tigecycline was calculated by fitting the dose-response curve using GraphPad Prism software. All experiments were performed in triplicate ($n = 3$ independent experiments).

## Purification of human 55S mitoribosomes

In all, 2 L of Expi293 cells were harvested by centrifugation at a density of $2 \times 10^6$/mL, washed in 1×PBS, and resuspended in 400 mL mitochondrial isolation buffer (50 mM HEPES-KOH, pH 7.5, 10 mM KCl, 1.5 mM $MgCl_2$, 1 mM EDTA, 1 mM EGTA, 1 mM DTT, and protease inhibitor). The resuspended cell pellet was lysed by grinding in a Dounce homogenizer. Lysate was clarified by centrifugation at $1000 \times g$ for 15 min at 4 °C, then centrifuged at $10,000 \times g$ for 15 min at 4 °C. The resulting pellet was resuspended in 10 mL resuspension buffer (50 mM HEPES-KOH, pH 7.5, 10 mM KCl, 1.5 mM $MgCl_2$, 1 mM EDTA, 1 mM EGTA, 1 mM DTT, 70 mM sucrose, 210 mM mannitol, and protease inhibitor). Genomic DNA was removed by the addition of 200 U RNase-free DNaseI and centrifugation again at $10,000 \times g$ for

15 min at 4 °C. The pellet was then resuspended in 2 mL resuspension buffer and loaded onto sucrose gradients (60%/32%/23%/15% sucrose in 20 mM HEPES-KOH, pH 7.5, 1 mM EDTA). The gradients were centrifuged at 139,000×$g$ for 1 h at 4 °C and the brown band containing mitochondria migrating to the interface of 32% and 60% sucrose was collected.

The collected mitochondria were resuspended in mitochondrial lysis buffer (25 mM HEPES-KOH, pH 7.5, 150 mM KCl, 50 mM Mg(OAc)$_2$, 2% Triton X-100, 2 mM DTT, and protease inhibitor). The mixture was homogenized with a glass dounce homogenizer. The lysate was cleared by centrifugation at 30,000 × $g$ for 20 min at 4 °C. The supernatant was then layered on top of sucrose cushions (34% sucrose in 20 mM HEPES-KOH, pH 7.5, 100 mM KCl, 20 mM Mg(OAc)$_2$, 1% Triton X-100, and 2 mM DTT) and centrifuged at 230,000 × $g$ for 45 min at 4 °C. The mitoribosomal pellets were resuspended in 100 μl resuspension buffer (20 mM HEPES-KOH, pH 7.5, 100 mM KCl, 20 mM Mg(OAc)$_2$, and 2 mM DTT).

The mitoribosomal suspension was layered on top of a sucrose density gradient (15–30% sucrose in 20 mM HEPES-KOH, pH 7.5, 100 mM KCl, 20 mM Mg(OAc)$_2$, and 2 mM DTT) and centrifuged at 260,800×$g$ (SW41Ti rotor) for 3 h at 4 °C. The gradients were subsequently fractionated from top to bottom using a Gradient Master (BioComp). The fractions corresponding to the 55S peak (monosome sample) were pooled and exchanged into buffer (20 mM HEPES-KOH, pH 7.5, 100 mM KCl, 20 mM Mg(OAc)$_2$, and 2 mM DTT). The sample was kept on ice until cryo-EM grid preparation.

## Yeast 80S ribosome purification
The cryo-EM sample and data of the tigecycline complex with the yeast 80S ribosome originate from a previous study and were described in detail before[33]. In brief, an *S. cerevisiae* strain with Not4-FtpA was cultured in 10 L YPD medium containing 5 μg/ml ampicillin and 10 μg/ml tetracycline. The cells were harvested at an OD600 of 0.9, washed in water and lysis buffer (20 mM HEPES pH 7.4, 100 mM KOAc, 10 mM Mg(OAc)$_2$, 1 mM DTT, 0.5 mM PMSF, 100 μg/ml tigecycline, protease inhibitor cocktail), and then frozen in liquid nitrogen. The frozen pellet was lysed by grinding, thawed in 15 ml of lysis buffer, and centrifuged at 12,000×$g$ for 15 min at 4 °C. The lysate was layered on 10–50% sucrose density gradients (prepared in 20 mM HEPES pH 7.4, 100 mM KOAc, 10 mM Mg(OAc)$_2$, 1 mM DTT, 10 μg/ml tigecycline) and centrifuged at 125,755×$g$ for 3 h at 4 °C. The gradients were fractionated, and fractions corresponding to the 80S peak were pooled. The pooled samples were incubated with IgG-coupled Dynabeads, washed with wash buffer (20 mM HEPES pH 7.4, 100 mM KOAc, 10 mM Mg(OAc)$_2$, 1 mM DTT, 10 μg/ml tigecycline), and eluted with AcTEV protease in elution buffer (20 mM HEPES pH 7.4, 100 mM KOAc, 10 mM Mg(OAc)$_2$, 1 mM DTT, 10 μg/ml tigecycline) for 1.5 h at 4 °C. Samples were kept on ice until cryo-EM grid preparation or frozen in liquid nitrogen and stored at −80 °C.

## Factor-free (empty) human 80S ribosome preparation
FreeStyle 293-F cells were grown to a density of $2.0 \times 10^6$ cells/mL in HyCell TransFx medium (Cytiva, SH30939) supplemented with 0.01% Poloxamer 188 (Sigma Aldrich, P5556), 1× penicillin–streptomycin (Gibco, 15140122) and 3× GlutaMAX (Gibco, 35050061), pelleted and washed with 1× PBS. The pellet was resuspended in HS Buffer (20 mM HEPES/KOH pH 7.5, 500 mM KOAc, 2 mM MgCl$_2$, 1 mM DTT) supplemented with c0mplete EDTA-free protease inhibitor (Roche, 4693132001) and subsequently centrifuged at 2182×$g$ for 15 min and 36,500×$g$ for 25 min. Ribosomes were pelleted from the cleared lysate through a high-salt sucrose cushion (20 mM HEPES/KOH pH 7.5, 500 mM KOAc, 2 mM MgCl$_2$, 1 mM DTT, 1 M sucrose) in a Type 70 Ti rotor (Beckman Coulter) at 106,700×$g$ for 18.5 h. To dissociate subunits the ribosome pellet was resuspended in HS Buffer. Nascent chains were released by incubation with 1 mM Puromycin on ice for

15 min and then at 37 °C for 10 min. 40S and 60S subunits were separated through a high-salt sucrose gradient (20 mM HEPES/KOH pH 7.5, 500 mM KOAc, 2 mM MgCl$_2$, 1 mM DTT, 10-40% sucrose) in a SW 40 Ti rotor (Beckman Coulter) at 284,600×$g$ for 2.5 h. Then subunits were pelleted through a low-salt sucrose cushion (20 mM HEPES/KOH pH 7.5, 100 mM KOAc, 10 mM MgCl$_2$, 1 mM DTT, 1 M sucrose) in a Type 70 Ti rotor (Beckman Coulter) at 140,900×$g$ for 14 h and resuspended in LS Buffer (20 mM HEPES/KOH pH 7.5, 100 mM KOAc, 10 mM MgCl$_2$, 1 mM DTT). Factor-free 80S were formed by incubation of equimolar amounts of 40S and 60S subunits on ice for 30 min. Then, 80S were separated from remaining non-associated subunits through a low-salt sucrose gradient (20 mM HEPES/KOH pH 7.5, 150 mM KOAc, 5 mM MgCl$_2$, 1 mM DTT) in a SW 40 Ti rotor (Beckman Coulter) at 284,600×$g$ for 2.5 h. The 80S fraction was pelleted through a low-salt sucrose cushion (20 mM HEPES/KOH pH 7.5, 100 mM KOAc, 10 mM MgCl$_2$, 1 mM DTT, 1 M sucrose) in a TLA-120.2 rotor (Beckman Coulter) at 434,513×$g$ for 1 h and resuspended in LS Buffer.

## Native human 80S ribosome purification
The human 80S ribosome was isolated using a modified protocol adapted from the yeast 80S ribosome purification process. In detail, 1 L of Expi293 cells were harvested by centrifugation, washed in 1X PBS, resuspended in 40 ml of lysis buffer (20 mM HEPES pH 7.4, 100 mM KOAc, 7.5 mM Mg(OAc)$_2$, 1 mM DTT, 100 μg/mL (-170 μM) tigecycline) and homogenized using a glass dounce homogenizer. The lysate was cleared by centrifugation at 10,000×$g$ for 15 min at 4 °C. The supernatant was then layered on top of six 10% to 40% sucrose density gradients in 50 mM HEPES pH 7.4, 500 mM KCl, 5 mM MgCl$_2$, 2 mM DTT, 10 μg/mL (-17 μM) tigecycline and 0.1 mM EDTA. The gradients were centrifuged at 260,800×$g$ (SW41Ti rotor) for 3 h at 4 °C. The fractions corresponding to the 80S peak (monosome sample) were pooled and exchanged into buffer (20 mM HEPES pH 7.4, 100 mM KOAc, 2.5 mM Mg(OAc)$_2$, 10 μg/mL (-17 μM) tigecycline and 2 mM DTT). The sample was kept on ice until cryo-EM grid preparation.

## Cryo-EM grids preparation
Factor-free/empty 80S ribosomes (140 nM) were incubated with either 4 μM or 100 μM tigecycline (-28.6- or 714.3-fold excess) for 45 min on ice before plunge freezing. 3.5 μL of sample was applied to precoated (3 nm carbon) R3/3 carbon support grids (Quantifoil) with 45 s pre-blotting and 3 s blotting time. Native human ribosomes (OD260 ≈ 5, -0.1 μM) were incubated with 100 μM tigecycline (-1000-fold excess) for 15 min on ice. Human mitoribosomes (OD260 ≈ 5, -0.16 μM) were incubated with 5 μM, 10 μM, and 100 μM tigecycline (approximately 31.25, 62.5, and 625-fold excess relative to the mitoribosome, respectively) for 15 min on ice. The latter samples (3.5 μL) were applied to precoated (2 nm) R1.2/1.3 carbon-supported copper grids (Quantifoil), blotted for 4 s at 4 °C, and plunge-frozen in liquid ethane using an FEI Vitrobot Mark IV.

## Electron microscopy and image processing
Data were collected on a Titan Krios cryo-electron microscope operating at 300 keV using EPU 2. Data for the yeast 80S ribosome were collected with a pixel size of 0.847 Å/pixel and within a defocus range of −0.8 to −2.5 μm using a K2 Summit direct electron detector under low dose conditions with a total dose of 44 e-/Å$^2$. Data for the empty human 80S ribosome were collected with a pixel size of 0.727 Å/pixel and within a defocus range of −0.8 to −2.5 μm using a Falcon 4i direct electron detector under low dose conditions with a total dose of 40 e-/Å$^2$. Data for the human 55S mitoribosome with 100 μM tigecycline were collected with a pixel size of 1.19 Å/pixel, while data for the native human 80S ribosome, the human 55S mitoribosome with 5 μM tigecycline, and the human 55S mitoribosome with 10 μM tigecycline were collected with a pixel size of 0.932 Å/pixel. These samples were collected within a defocus range of −1 to −2.5 μm using a Falcon IV

direct electron detector under low dose conditions with a total dose of 50 e-/Å[2]. The original image stacks were dose-weighted, aligned, summed, and drift-corrected using MotionCor2[40]. Contrast-transfer function (CTF) parameters and resolutions were estimated for each micrograph using CTFFIND4 and GCTF, respectively[41,42]. Micrographs with an estimated resolution of less than 5 Å and an astigmatism of less than 5% were manually inspected for contamination or carbon rupture.

For the dataset collected from the yeast ribosome sample (the tigecycline-stalled monosome sample), a total of 11,217 good micrographs were selected. Automatic particle picking was performed in Gautomatch v0.56 without reference, yielding a total of 1,116,932 particles. Particle extraction was performed in Relion 3.1[43], but the subsequent 2D classification was performed in cryoSPARC v3.2[44]. After removing all junk classes, only those classes were selected that had a good representation of 80S ribosomes. In the end, a total of 307,386 particles were selected and sent back to Relion 3.1[43]. 3D refinement and unaligned 3D classification were performed to obtain the desired classes. Two classes of yeast ribosomes could be classified: one containing eEF2 and eIF5A, and the other containing P-site initial tRNA and Not5[33]. The second class was selected and refined to the final map after CTF refinement in Relion 3.1. The detailed procedure is illustrated in Supplementary Fig. 10.

For the human 55S mitoribosome with 100 µM tigecycline dataset, a total of 6,663 good micrographs were selected. The same procedure was followed as for the yeast 80S ribosome dataset. First, a total of 1,204,711 particles were selected in Gautomatch v0.56 without using any reference. The particles were then extracted in Relion 3.1[43] and imported into cryoSPARC v3.2[44] where 2D classification was performed. As a result, only a total of 910,845 good particles with a clear 55S/39S mitoribosome shape were selected and exported back to Relion 3.1[43]. 3D refinement and 3D classification were performed to obtain the desired classes. In the end, most of the particles showed bad 28S small subunit, so they were discarded. Only one class with a very nice 55S mitoribosome structure was selected. In the case of 39S mitoribosome, as we found before, most resemble the biogenesis intermediates[45]. Only a class containing a set of 12,156 particles is the mature 39S mitoribosome. CTF refinement was performed before we obtain the final maps. The detailed procedure is illustrated in Supplementary Fig. 2.

For the dataset collected from the human 80S ribosome, a total of 7673 good micrographs were selected. Automatic particle picking was done in Gautomatch v0.56, resulting in a total number of 542,730 particles. The particles were extracted and imported into cryoSPARC v3.2[44]. After 2D classification, a number of 380,031 good particles were directly selected for 3D classification with alignment in cryoSPARC v3.2[44], which resulted in six classes, four of which resembled nice human 80S ribosome structures, so these four classes were collected and imported into Relion 3.1[43] for further 3D classification. As a result, four different classes of 80S ribosomes could be sorted out: the first one contains E-tRNA and CCDC124 in classical conformation with both the 40S body and head in non-rotated state[46]; the second one contains E-site tRNA, P-tRNA and mRNA in hybrid conformation with the 40S body rotated CCW (counterclockwise); the third one contains E-tRNA and CCDC124 in hybrid state with only the 40S head rotated CCW; the last one contains E-tRNA, eEF2 and SERBP1 representing the classical hibernating 80S ribosome[46]. All four classes were selected, and 3D refinement and CTF refinement were applied in Relion 3.1 to get the final reconstructions. The detailed procedure is illustrated in Supplementary Fig. 7.

For the human 55S mitoribosome with 5 µM tigecycline dataset, 6,327 high-quality micrographs were selected. Subsequently, a total of 1,585,892 particles were selected in Gautomatch v0.56 without using any reference. The particles were then extracted in Relion 3.1[43] and imported into cryoSPARC v3.2[44]. The imported particles were directly subjected into heterogeneous refinement. As a result, a total of 371,714

particles represented human 55S mitoribosome, while 622,396 particles represented the immature 39S mitoribosome. The class corresponding to the human 55S mitoribosome was selected and exported back to Relion 3.1[43]. Two rounds of 3D classification were performed to obtain the desired classes. In the end, one class (122,982 particles) exhibited a well-defined 55S mitoribosome structure, and another class (57,266 particles) displayed a well-defined 39S mitoribosome structure. All remaining classes, whether representing low resolution or bias, were discarded. Finally, CTF refinement was performed to obtain the final maps. The detailed procedure is illustrated in Supplementary Fig. 3.

For the human 55S mitoribosome with 10 µM tigecycline dataset, the same procedure was applied as used for the human 55S mitoribosome with 5 µM tigecycline dataset. In brief, 1,493,828 particles were picked from 5815 high-quality micrographs using Gautomatch v0.56. After heterogeneous refinement in cryoSPARC v3.2[44], a total of 291,791 particles representing a well-defined human 55S mitoribosome were selected and imported into Relion 3.1[43]. Following two rounds of 3D classification, one class (83,274 particles) representing the 55S mitoribosome and another one (52,961 particles) representing the 39S mitoribosome were selected and refined to 3.3 Å and 3.1 Å, respectively. The detailed procedure is illustrated in Supplementary Fig. 3.

For the human empty 80S ribosome with 4 µM tigecycline dataset, a total of 591,152 particles were picked from 10,776 high-quality micrographs using Gautomatch v0.56. After particle extraction in Relion 3.1[43], particles were then directly imported into cryoSPARC v3.2[44] for heterogeneous refinement. A total of 209,408 particles representing a well-defined human 80S ribosome were selected and imported back into Relion 3.1[43]. Following one round of 3D classification, one class (98,844 particles) representing the 80S ribosome in high resolution was selected and refined to 2.2 Å using cryoSPARC v3.2. The detailed procedure is illustrated in Supplementary Fig. 6.

For the human empty 80S ribosome with 100 µM tigecycline dataset, the identical procedure was applied as used for the human empty 80S ribosome with 100 µM tigecycline dataset. In brief, a total of 1,377,458 particles were picked from 22,957 high-quality micrographs using Gautomatch v0.56. Following the processing in both Relion 3.1[43] and cryoSPARC v3.2[44], in the end, one class (310,531 particles) representing the 80S ribosome in high resolution was selected and refined to 2.0 Å using cryoSPARC v3.2. The detailed procedure is illustrated in Supplementary Fig. 6.

During processing, cryoSPARC v3.2[44] and Relion 3.1[43] automatically generated sphere masks were consistently used unless otherwise specified. To obtain the final high-resolution reconstructions for all the structures, automatically generated ribosome masks in Relion 3.1[43], using ribosomes (55S, 39S, and 80S ribosomes) as inputs, were applied.

## Atomic model building and refinement

In general, the structures of the yeast 80S ribosome (PDB: 6Z6J, 6TB3)[33,46], the human 80S ribosome (PDB: 6Z6L, 6Z6M)[46] and the human 55S mitoribosome (PDB: 7A5I)[47] were used for rigid body fitting into the respective maps, with some manual adjustments in Coot v0.9[48]. The tigecycline molecule was fetched from the Coot ligand library using the three-letter code: T1C and manually adjusted in Coot v0.9[48] according to the high-resolution maps. Ribosomal regions (such as Tig2 binding region in 55S mitoribosome) that changed after tigecycline binding were also manually adjusted. In the structure of the human 80S ribosome with tigecycline, E-tRNA, P-tRNA, and mRNA, the sequence of the mRNA is unknown. Therefore, the tRNA and mRNA in this model are used as placeholders.

The final models were real-space refined with secondary structure restraints using the PHENIX suite v1.19[49]. The restraint for the ligand tigecycline was automatically generated with Phenix.ready_set[49]. Final model evaluation was performed using MolProbity[50].

## Figure preparation

All figures showing cryo-EM maps and the molecular model were generated in ChimreaX v1.6[51]. The final refinement of the cryo-EM maps is done in Relion 3.1[43], and to prepare the figures all maps were either filtered in Relion 3.1 according to their local resolution estimation or filtered with DeepEMhancer v0.3[52]. All plots were created using GraphPad Prism 8. Gray scale calculation of protein bands in Fig. 1 was performed using Image Lab 6.1(Bio-Rad) and ImageJ1[53].

## Reporting summary

Further information on research design is available in the Nature Portfolio Reporting Summary linked to this article.

## Data availability

All cryo-EM maps and molecular models generated in this study have been deposited in the Electron Microscopy Data Bank (EMDB) and in the Protein Data Bank (PDB) with accession codes: EMD-36836 and 8K2A for the 55S mitoribosome + 100 μM tigecycline; EMD-36837 and 8K2B for the 39S mitoribosome + 100 μM tigecycline; EMD-36838 and 8K2C for the human 80S ribosome with tigecycline, E-tRNA and CCDC124; EMD-36839 and 8K2D for the yeast 80S ribosome with tigecycline, eEF2, Stm1 and eIF5A; EMD-36945 and 8K82 for the yeast 80S ribosome with tigecycline, Not5 and P-tRNA; EMD-38632 and 8XT0 for the 55S mitoribosome + 5 μM tigecycline; EMD-38633 and 8XT1 for the 39S mitoribosome + 5 μM tigecycline; EMD-38634 and 8XT2 for the 55S mitoribosome + 10 μM tigecycline; EMD-38635 and 8XT3 for the 39S mitoribosome + 10 μM tigecycline; EMD-38629 and 8XSX for the human 80S ribosome with tigecycline, E-tRNA, SERBP1 and eEF2; EMD-38630 and 8XSY for the human 80S ribosome with tigecycline, E-tRNA and CCDC124 (40S head swiveled); EMD-38631 and 8XSZ for the human 80S ribosome with tigecycline, E-tRNA, P-tRNA and mRNA; EMD-39455 and 8YOO for the human empty 80S ribosome with 100 μM tigecycline; EMD-39456 and 8YOP for the human empty 80S ribosome with 4 μM tigecycline. Source data are provided with this paper.

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

## Acknowledgements

The authors thank the Center of Cryo-Electron Microscopy, Core Facility of Shanghai Medical College, Fudan University for technical support. This research was supported by grants from the National Key Research and Development Program of China (2023YFC2413200/2023YFC2413204), the National Natural Science Foundation of China (32371350), the Shanghai Municipal Science and Technology Commission grants (22ZR1413600, 22410712400) to J.C., and by a European Research Council (ERC) Advanced Grant (ADG 885711 HumanRibogenesis) to R.Be.

## Author contributions

X.L., M.W., T.D., R.Be., and J.C. conceived the study. X.L., M.W., and R.Bu. prepared the samples for cryo-EM. M.W. and T. D. characterized the inhibition of tigecycline in vivo and in vitro. Y.L. collected cryo-EM data. J.C. processed the data and built and refined the models. J.C. and R.Be. analyzed and interpreted the structures. X.L., M.W., T.D., R.Be., and J.C. wrote the manuscript. All authors commented on the manuscript.

## Competing interests

The authors declare no competing interests.
