## [Peer Review File · Nature Communications]

Structural basis for differential inhibition of eukaryotic ribosomes by tigecyclineReviewer #1 (Remarks to the Author):

In the present paper, Li and colleagues aim to uncover the mechanism by which the antibiotic tigecycline inhibits eukaryotic translation. To this extent, they present cryo-EM structures of tigecycline bound to a variety of eukaryotic ribosomes. The inhibitory mechanism of tigecycline for bacterial translation is quite well understood. However, the action of the antibiotic on eukaryotic ribosomes is important to understand side effects. Moreover, tigecycline has anti-inflammatory and anti-tumorigenic activity. Thus, the study could be potentially important.

Unfortunately, the interpretation of the results is confounded by a variety of problems. First of all, to infer the mechanisms of inhibition based on structural results alone is highly dangerous. The structural analysis can make only predictions that have then to be tested from a functional perspective. Accordingly, the authors have to perform functional assays to demonstrate inhibition of PTC and translocation for the mitoribosome directly.

Preparation of the complexes appears as a second major problem, as the authors have used a concentration of 100 μ M tigecycline. This is about two orders of magnitude higher than the IC50 they report for the mitochondrial system. This calls into question whether the observed binding sites are occupied at all at physiological concentration. The authors have to repeat the structural analysis of the mitoribosome at a physiological relevant tigecycline concentration.

The unsystematic selection of samples makes a meaningful comparison difficult. A crude preparation of human 80S ribosomes was used for analysis that contains additional factors. The authors chose a complex containing the hibernation factor CCDC124 for discussion, as it "contains all tigecycline binding sites observed in other states". Do the other complexes have all binding sites as well? Why the structure in Extended Data Fig.5 contains an E-site tRNA? This is not mentioned in the main text. How relevant is tigecycline binding to hibernating ribosomes? For the yeast 80S ribosome the authors chose a complex with Not5 and P-site tRNA. The absence of Tig3 and Tig4 (observed for the human 80S with E-site tRNA) in the absence of E-site tRNA implies that ribosomal ligands may be important for tigecycline binding and calls for a more systematic comparison. Furthermore, detailed conclusions with respect to a mechanism of inhibition in the absence of additional data are not warranted.

It is unclear what the mechanism of tigecycline inhibition has to do with the presence of methionine tRNA (page 15, Extended Data Fig.8). Human 80S were purified in the absence of the drug.

Why the data from the 2020 Buschauer et al. paper needed to be reprocessed? Is tigecycline not present in the original cryo-EM structures? Furthermore, it is curious that a major class from the 2020 paper are ribosomes was reported to contain eIF5A. Now the major class besides the Not5 complex has eIF5A, eEF2 and Stm1. This discrepancy has to be explained.

Reviewer #2 (Remarks to the Author):

Using cryo-electron microscopy, Li et al. determined the structure of the human mitoribosome in complex with tigecycline - a broad-spectrum antibiotic commonly used for treating multiple resistant bacteria. The structure reveals four distinct binding sites for tigecycline within the mitoribosome. The authors also determined the structure of the human 80S ribosome in complex with tigecycline, which reveals several binding sites for this antibiotic. These structures collectively contribute to our understanding of how tigecycline exerts its toxicity, shedding light on its molecular mechanisms. This manuscript presents some novel findings that should be published after a revision.

Figure 1: the authors should ensure the representation of every measurement point with its corresponding error bar. This would provide a more comprehensive and accurate visualization of the data, facilitating a clearer understanding of the precision and variability associated with each

measurement.

The authors mentioned in the manuscript that, "Although the tigecycline molecule (Tig1) on the 39S mitoribosome was not at high resolution, it was still sufficiently resolved for an unambiguous assignment (Fig. 2b)." Despite the lack of explicit evidence in Figure 2b to support the resolution-based assignment, the observed binding site on the 55S mitoribosome strongly suggests that this density corresponds to a tigecycline molecule. The authors should improve the segmentation of the map in this region and provide a clear model-to-map fitting representation. Additionally, it might be beneficial for the authors to include a discussion sentence acknowledging that the assignment of this density on the 39S mitoribosome was based on the observed binding site on the 55S. Suggestions also extend to improving the overall clarity of Figure 2.

Jenner et al., 2013 (reference 11 in this manuscript) determined the X-ray crystallography Structure of the 70S-tigecycline complex, which revealed the primary binding site. The authors identified four distinct binding sites in this study - despite the potential conservation between bacterial and human mitochondrial ribosomes. Can they further comment on this discrepancy?

In the X-ray structure, the ratio between tigecycline and ribosome was at a 20-fold excess, while here, it is unclear what ratio was used for the cryo-EM data. The authors may consider disclosing the ratio used in their experimental conditions.

One should also inquire whether the authors would detect these secondary binding sites at concentrations of tigecycline that are clinically relevant. Addressing this point would provide valuable insights into the practical implications of the identified binding sites and their potential relevance in the toxicity of tigecycline.

Figure 1 presents the concentration-dependent inhibition of the 80S ribosome by tigecycline. Although the inhibition appears mild, the structure reveals the presence of six molecules of tigecycline bound to the 80S ribosome. Can this discrepancy be explained by the concentration of the antibiotic used for cryo-EM? The authors used a 1000-fold excess of tigecycline compared to the ribosome (approximately 0.1 μ M vs. 100 μ M). This point needs further discussion in the manuscript. The manuscript presently lacks clarity regarding the specificity of these interactions.

Point-by-point-response

Reviewer #1 (Remarks to the Author):

In the present paper, Li and colleagues aim to uncover the mechanism by which the antibiotic tigecycline inhibits eukaryotic translation. To this extent, they present cryo-EM structures of tigecycline bound to a variety of eukaryotic ribosomes. The inhibitory mechanism of tigecycline for bacterial translation is quite well understood. However, the action of the antibiotic on eukaryotic ribosomes is important to understand side effects. Moreover, tigecycline has anti-inflammatory and anti-tumorigenic activity. Thus, the study could be potentially important.

A: Thank you for this very positive comment.

Unfortunately, the interpretation of the results is confounded by a variety of problems. First of all, to infer the mechanisms of inhibition based on structural results alone is highly dangerous. The structural analysis can make only predictions that have then to be tested from a functional perspective. Accordingly, the authors have to perform functional assays to demonstrate inhibition of PTC and translocation for the mitoribosome directly.

A: We agree that the inhibition of the PTC or tRNA translocation cannot be directly deduced from a purely structural perspective. To emphasize the structural focus of our paper, we have rephrased the title to “**Structural basis of inhibition of eukaryotic ribosomes by tigecycline**”, as well as the chapter headlines and other parts of the text accordingly. Together, we aim to make it clear that our conclusions are derived from our structural data in consideration of previously published structural and biochemical work.

While the suggested functional assays are desirable, it is not currently possible to perform these highly demanding experiments, which would be necessary to provide more definitive answers. Although *in vitro* translation systems, such as the PURE system, are readily available for *E. coli*, no such systems exist for human mitochondrial translation. Newly establishing an equivalent human mitochondrial system in order to perform similar functional experiments to carefully dissect the potential mechanisms would require an excessive amount of time and effort that is unfortunately beyond the capabilities of this revision.

However, we feel that we can make plausible suggestions when taking into account our structural findings together with published observations. For example, we find that one novel binding site of tigecycline (Tig1) in the human mitoribosome PTC is identical to the TcmX binding site on *E. coli* 70S ribosomes (see main text). Only recently, it has been shown structurally and biochemically that TcmX selectively impairs peptidyl transfer when encountering QK motifs in the nascent polypeptide chain in *E. coli* (<https://www.nature.com/articles/s41589-023-01343-0>, Ref. 28 in main text), mainly using a combination of *in vitro* translation, next-generation sequencing and cryo-EM. This allows us to suggest that tigecycline (Tig1) may employ a similar mechanism to block protein synthesis on human mitoribosomes.

Moreover, we proposed an additional potential translation inhibition mechanism by tigecycline (Tig2) and carefully adjusted the wording in order to avoid overinterpretation of our structural data. Again, we agree that blocking translocation of the tRNA in the P-site cannot be directly concluded from structures. However, for Tig2 we observed a substantial conformational rearrangement of nucleobases upon Tig2 binding on the 55S mitoribosome (Fig. 3) being highly suggestive for a potential inhibitory effect. We feel this provides a valuable hypothesis for future biochemical mechanistic studies.

We have now revised our conclusion as follows:

“Taken together, Tig1 and Tig2 are likely to impair the mitoribosomal PTC and, in addition, to prevent tRNA translocation.”

We also added this aspect to our revised Discussion to point out the limitation of our study:

“It is important to note that our structural analysis provides predictions of the inhibition mechanism, but further functional studies are necessary to validate these hypotheses.”

Preparation of the complexes appears as a second major problem, as the authors have used a concentration of 100 μM tigecycline. This is about two orders of magnitude higher than the IC_{50} they report for the mitochondrial system. This calls into question whether the observed binding sites are occupied at all at physiological concentration. The authors have to repeat the structural analysis of the mitoribosome at a physiological relevant tigecycline concentration.

A: We appreciate the valuable feedback concerning the tigecycline concentration used in our study. The concentration of 100 μM was selected based on considerations of achieving robust structural data and capturing the binding

interactions effectively. However, we recognize the importance of conducting experiments at physiologically relevant concentrations to enhance the clinical significance of our findings.

In response to this concern, we made an effort to perform additional experiments using lower tigecycline concentrations that align more closely with physiological conditions. Although there is not a very consistent physiological concentration of tigecycline in our body after the treatment, numerous studies (e.g., references below) suggest an average serum concentration ranging from 4 to 10 μM . Notably, certain tissues exhibit significantly higher concentrations, such as an 8.6-fold increase in the lung, a 38-fold increase in the gall bladder, a 78-fold increase in alveolar cells, where tigecycline concentrations can accumulate to several hundred μM (see references below).

We have performed two additional structural analysis of the human mitoribosome at tigecycline concentrations of 5 μM and 10 μM . In these new structures, Tig1, Tig2, and Tig3 are clearly observed in the 55S mitoribosomes treated with both 5 μM and 10 μM tigecycline. Yet, the presence of Tig4 is not in these new structures, suggesting its functional relevance primarily under high tigecycline concentrations. We also specify this in the following description of the Tig4 molecule. We have incorporated these findings into our revised manuscript (Figure 2, Results sections 2, 4, Discussion and supplementary figure 5) and discussed the observed differences as followed:

Results sections 2:

“In contrast, when the human mitoribosomes were exposed to 5 μM or 10 μM tigecycline, concentrations within the physiological range during clinical usage²³⁻²⁵, only three tigecycline molecules (Tig1-3) could be distinctly assigned, with Tig4 absent (Fig. 2a, b; Supplementary Fig. 5a). This discrepancy suggests that the binding of Tig4 occurs specifically under high concentrations of tigecycline.”

Results sections 4:

“Tig4 binds to a new region near h1 and h44 of the 12S rRNA (Fig. 4d), exclusively observed in the 55S mitoribosome treated with 100 μM tigecycline.”

Discussion:

“The Tig4 molecule on the other hand is only detected in the human 55S mitoribosome treated with 100 μM tigecycline, suggesting that Tig4 binds specifically under high concentrations of tigecycline.”

These additional structural analyses validate our observations under conditions that better simulate the *in vivo* environment, providing a more comprehensive understanding of tigecycline binding and its physiological implications.

Reference:

1. https://www.accessdata.fda.gov/drugsatfda_docs/nda/2005/21-821_Tygacil.cfm
2. Rodvold, K. A. et al. Serum, tissue and body fluid concentrations of tigecycline after a single 100 mg dose. *J Antimicrob Chemother* 58, 1221-1229, doi:10.1093/jac/dkl403 (2006).
3. Meagher, A. K., Ambrose, P. G., Grasela, T. H. & Ellis-Grosse, E. J. The pharmacokinetic and pharmacodynamic profile of tigecycline. *Clin Infect Dis*, 41, Suppl 5, S333-340, doi:10.1086/431674 (2005).
4. Brink, A. J. et al. Guideline: appropriate use of tigecycline. *S Afr Med J* 100, 388-394, doi:10.7196/samj.4109 (2010).

The unsystematic selection of samples makes a meaningful comparison difficult. A crude preparation of human 80S ribosomes was used for analysis that contains additional factors. The authors chose a complex containing the hibernation factor CCDC124 for discussion, as it “contains all tigecycline binding sites observed in other states”. Do the other complexes have all binding sites as well?

A: In response to this concern, we would like to clarify our methodology: Traditionally, when studying antibiotic mechanisms inhibiting translation, empty ribosomes (70S or 80S) were initially purified and then incubated with antibiotic molecules in solution. However, for our investigation into the impact of tigecycline on human cytoplasmic translation, we adopted a different approach. We pre-treat the human cells with tigecycline before cell lysis. Subsequently, we did the crude preparation of the human 80S ribosomes. This approach was chosen, since it could purify distinct physiological/functional ribosome complexes with tigecycline that better reveal potential mechanisms underlying tigecycline’s inhibition of specific steps of translation. Additionally, it allowed us to make a valid comparison with a similarly tigecycline-treated yeast 80S sample from our previous publication. Although, the high concentrations (170 μM) used for treatment for the native human and yeast ribosome preparations are not clinically/physiologically relevant but allow for stabilization of cytoplasmic translation intermediates and have been used for this purpose before (see references #33 and #34 in the manuscript). Providing structural insights into the generated intermediates appears valuable for the scientific community and therefore, following a similar preparation protocol was explicitly chosen.

To address this concern in the manuscript, we now provide a more systematic and comprehensive analysis by including all human 80S ribosome complexes in our revised manuscript side by side with the yeast 80S complexes, including:

- 1) State human 80S+Tig (with E-tRNA and CCDC124): The structure previously discussed in our manuscript.
- 2) State human 80S+Tig (with E-tRNA and CCDC124, 40S head swiveled): This structure contains CCDC124, but the 40S head is rotated (counterclockwise) compared to the previous state.
- 3) State human 80S+Tig (E-tRNA, eEF2, and SERBP1): A state containing eEF2 and SERBP1.
- 4) State human 80S+Tig (with E-tRNA, P-tRNA, and mRNA): An active translating 80S ribosome.
- 5) State yeast 80S + Tig (with eEF2, Stm1 and eIF5a): A state containing eEF2 and Stm1.
- 6) State yeast 80S + Tig (with Not5, P-tRNA and mRNA): A paused 80S ribosome with quality control factor Not5

For our four different structures of the native human 80S ribosomes and two of native yeast 80S ribosomes with different numbers of tigecycline molecules, differences are now illustrated in a new Figure 6 and in new/revised Supplementary Figures 7-11 and Supplementary Tables 2-3. We further provide detailed descriptions in the main text with a new and revised chapter “**Tigecycline binds to multiple sites on the native eukaryotic 80S ribosomes**”. Specifically, we clarify the presence or absence of tigecycline molecules in each state, explaining the variations observed and highlighting common binding sites, as following:

“To further explore the effect of tigecycline on cytoplasmic translation, we purified native 80S ribosome complexes from human HEK293 cells treated with a high dose of tigecycline (170 μ M). Using cryo-EM single particle analysis and extensive 3D sorting, we obtained four different ribosome structures with different numbers of bound tigecycline molecules (Fig. 6; Supplementary Figs. 7-8; Supplementary Tables 2-3): 1) the human 80S+Tig (with E-tRNA and CCDC124) state; 2) the human 80S+Tig (with E-tRNA and CCDC124, 40S head swiveled) state; 3) the human 80S+Tig (E-tRNA, eEF2, and SERBP1) state; and 4) the human 80S+Tig (with E-tRNA, P-tRNA, and mRNA) state. These structures represent different translational states, characterized by different rotations of the 40S ribosome body or head, accompanied by different translation factors (Fig. 6a).

In the human 80S+Tig (with E-tRNA and CCDC124) state, we identified six tigecycline binding sites housing a total of nine tigecycline molecules (Tig1-10, except Tig3) on the human 80S ribosome (Fig. 6b). Notably, most of these molecules were located in the non-conserved peripheral region, where translation is less likely to be affected (Fig. 6b; Supplementary Fig. 9; Supplementary Table 3). Tig10, binding to the E-tRNA and the 40S head, was exclusive to this state and absent in others (Fig. 6b; Supplementary Fig. 9d, e; Supplementary Table 3). Conversely, Tig9, also associated with E-tRNA in the both human 80S+Tig (with E-tRNA and CCDC124) and human 80S+Tig (E-tRNA, eEF2, and SERBP1) states (Fig. 6b; Supplementary Fig. 9b, c; Supplementary Table 3). Speculatively, Tig9-10 represent E-site tRNA and 40S head conformation-dependent binding sites.

In the human 80S+Tig (with E-tRNA, P-tRNA, and mRNA) state, representing an actively translating 80S ribosome, only the tigecycline molecules (Tig4-8) bound to the peripheral region persisted (Fig. 6b; Supplementary Fig. 9f-h; Supplementary Table 3), suggesting that these sites do not affect translation. The absence of Tig1-3 molecules in this actively translating ribosome implies that Tig1-3 may exert only a mild influence on human translation (Supplementary Table 3), consistent with our analysis of tigecycline with empty 80S ribosomes (Fig. 5).

To validate these findings on the human 80S ribosome, we also solved the cryo-EM structure of the tigecycline complex with the yeast 80S ribosome from one of our published datasets³³ (Fig. 6; Supplementary Fig. 10; Supplementary Tables 2-3), where a similar high-dose treatment (170 μ M) was used. Unlike the reported structures, we selected the dataset derived from the tigecycline-stalled monosome sample³³. We obtained two different high-resolution yeast 80S ribosome structures, one state with eEF2/Stm1/eIF5A binding and the other state with Not5 and P-site tRNA binding.

We found four tigecycline binding sites housing a total of six tigecycline molecules on the yeast 80S ribosome (Fig. 6; Supplementary Fig. 11; Supplementary Tables 2-3). Interestingly, these six tigecycline molecules (Tig1-3, 11-13) also form dimers and trimers, and are mostly located at the peripheral region of the yeast 80S ribosome (Fig. 6; Supplementary Fig. 11). Consistent with a previous biochemical study³⁴ and our analysis of the human 80S ribosome (Fig. 5), we find a tigecycline molecule (Tig3) binding in the A-site (Supplementary Fig. 11c) in the yeast 80S+Tig (with Not5, P-tRNA and mRNA) state, which explains its canonical tRNA accommodation inhibition activity³⁴. Furthermore, we find that in both human and yeast 80S ribosomes, the associated tRNA is the initial methionine tRNA (Supplementary Fig. 12), which could be a result from the Tig3 molecule blocking the incoming A-site tRNA after initiation¹⁴.

Moreover, the additional binding site of Tig1-2 observed in the yeast 80S+Tig (with Not5, P-tRNA and mRNA) state or Tig1-3 in the yeast 80S+Tig (with eEF2, Stm1, eIF5a) state is conserved compared to the binding site of Tig1-2 on

the human ribosome, which is located near the L1 stalk and the E-site tRNA (Fig. 5d; Supplementary Fig. 11d). This suggests that high concentration of tigecycline might inhibit eukaryotic 80S ribosomes through two conserved binding sites: one is the canonical “primary” binding site of Tig3, and the other is the L1 stalk region of the Tig1-2 molecules.”

Furthermore, in another attempt to improve this manuscript we prepared empty/factor-free human 80S ribosomes with both physiological and high concentrations of tigecycline (4 μ M and 100 μ M) and performed cryo-EM single particle analysis. This provides a more meaningful comparison with our mitoribosome data. It also shows more clearly the difference in tigecycline binding: only at higher concentrations tigecycline binds to cytoplasmic ribosomes while binding to 3 out of 4 sites (Tig1-3) on mitoribosomes already occurs at clinically relevant concentrations. Accordingly, this data is described and illustrated in a new chapter “Tigecycline does not bind to human 80S ribosomes at physiological conditions” and illustrated in a new Figure 5 and Supplementary Figure 6.

Why the structure in Extended Data Fig.5 contains an E-site tRNA? This is not mentioned in the main text.

A: We apologize for this mistake. E-site tRNA exists in all four human 80S ribosome structures. We now mentioned it in our revised main text.

How relevant is tigecycline binding to hibernating ribosomes?

A: Thank you for pointing this out. Hibernating ribosomes are a result of an important protective mechanism of cells to tune down translation in response to stress (including antibiotics treatment). By isolating 80S from tigecycline treated cells we aimed to capture ribosomes as close as possible to their native tigecycline bound states. Here, we observe most tigecycline molecules bound to human 80S ribosomes (at 6 sites) in one of these states (human 80S + Tig with CCDC124, Fig 6, Supplementary Figs 7 and 8). As described in the main text, we initially focused on this state to describe all observed binding sites, and further noted differences to the actively translating or other states. However, we think that speculations about whether it effects hibernating ribosomes as such are not warranted based on our structural data alone. Meanwhile, we also cannot confidently reason whether they occur as a result of tigecycline treatment.

For the yeast 80S ribosome the authors chose a complex with Not5 and P-site tRNA. The absence of Tig3 and Tig4 (observed for the human 80S with E-site tRNA) in the absence of E-site tRNA implies that ribosomal ligands may be important for tigecycline binding and calls for a more systematic comparison. Furthermore, detailed conclusions with respect to a mechanism of inhibition in the absence of additional data are not warranted.

A: Thank you for pointing this out. As described above, the cytoplasmic complexes generated by high-dose treatment of human and yeast cells are natively occurring complexes. We chose the major classes that are isolated upon tigecycline treatment, whether yeast and human cytoplasmic ribosomes respond differently to the treatment or whether there is a difference in the natural abundance of translational intermediates already before treatment are possibilities we indeed cannot speculate about. Nonetheless, we provide valuable structural insights for researchers regarding tigecycline binding when used at high concentrations to target cytoplasmatic translation.

To address the question:

Firstly, as stated above, we now provide a more in-depth comparison between native human and yeast 80S ribosomes structures (new revised paragraph, Fig. 6, Supplementary Fig.7-11 and Supplementary Tables 2-3).

Secondly, we now address the potential effect of E-site tRNA or translation factors for tigecycline in our discussion (we renamed Tig3 and Tig4 to Tig9 and Tig10):

“Furthermore, in human 80S ribosomes, two other binding sites, Tig9 and Tig10, interact either directly with E-site tRNA or very close to the anticodon stem, which could also contribute to inhibition (Supplementary Fig. 10b-d) at certain translation stages. Contrary, we do not observe E-site tRNA in our yeast 80S ribosome. Therefore, we cannot confirm the conservation of these two new sites. These observations raise the possibility that tigecycline could inhibit the ribosome during specific translational states or in dependence on particular translation factors, characteristics that may be shared with other antibiotics. Therefore, this could herald a novel strategy for antibiotic design.”

Thirdly, we agree that structural data alone is not sufficient to make definitive conclusion about inhibition mechanisms. Accordingly, we rephrased the discussion:

“This suggests that high concentration of tigecycline might inhibit eukaryotic 80S ribosomes through two conserved binding sites: one is the canonical “primary” binding site by Tig3, and the other is the L1 stalk region by Tig1-2 molecules.”

Lastly, in the discussion, to stress the structure-based nature of our findings:

“It is important to note that our structural analysis provides predictions of the inhibition mechanism, but further functional studies are necessary to validate these hypotheses.”

It is unclear what the mechanism of tigecycline inhibition has to do with the presence of methionine tRNA (page 15, Extended Data Fig.8). Human 80S were purified in the absence of the drug.

A: We are sincerely sorry for this mistake in properly describing the approach. We actually purified the human 80S following the same protocol as for the yeast sample. We also treated the human cells with tigecycline before cell lysis. We now corrected this in the revised methods.

Why the data from the 2020 Buschauer et al. paper needed to be reprocessed?

A: We apologize for the confusion. 1) We previously focused on different aspects of the same dataset, specifically the Not5 protein. Therefore, we used a focused mask, restricting classification to the E-site tRNA area. In our current analysis, with a specific interest in tigecycline, we have applied a general 3D classification approach to distinguish various conformational states. 2) Additionally, due to the development of the cryo-EM in these years, we want to take the advantage of the new software to improved structural interpretation, such as new version of Relion 3.1.4 and DeepEMhancer.

Is tigecycline not present in the original cryo-EM structures?

A: The tigecycline density is present in the original cryo-EM structures.

Furthermore, it is curious that a major class from the 2020 paper are ribosomes was reported to contain eIF5A. Now the major class besides the Not5 complex has eIF5A, eEF2 and Stm1. This discrepancy has to be explained.

A: We apologize for any confusion. In the 2020 paper, the authors originally collected four different datasets. For the current study, we have reprocessed the dataset collected from the tigecycline-stalled monosome sample (Fig. S4A in Buschauer *et al.*, Science, 2020). In this reprocessed dataset, the major class, aside from the one with Not5, includes eIF5A, eEF2, and Stm1. It's important to note that, in the 2020 paper, the class containing eIF5A, A-tRNA, and P-tRNA was derived from the cycloheximide-stalled monosome sample (Fig. S4C in Buschauer *et al.*, Science, 2020). We now carefully clarify this in our revised manuscript and methods part.

In the main text:

“To validate these findings on the human 80S ribosome, we also solved the cryo-EM structure of the tigecycline complex with the yeast 80S ribosome from one of our published datasets³³ (Fig. 6; Supplementary Fig. 11; Supplementary Tables 2-3), where a similar high-dose treatment (170 μ M) was used. Unlike the reported structures, we selected the dataset derived from the tigecycline-stalled monosome sample³³.”

In the methods:

“For the data set collected from the yeast ribosome sample (the tigecycline stalled monosome sample), a total of 11,217 good micrographs were selected.”

Reviewer #2 (Remarks to the Author):

Using cryo-electron microscopy, Li et al. determined the structure of the human mitoribosome in complex with tigecycline - a broad-spectrum antibiotic commonly used for treating multiple resistant bacteria. The structure reveals four distinct binding sites for tigecycline within the mitoribosome. The authors also determined the structure of the human 80S ribosome in complex with tigecycline, which reveals several binding sites for this antibiotic. These structures collectively contribute to our understanding of how tigecycline exerts its toxicity, shedding light on its molecular mechanisms. This manuscript presents some novel findings that should be published after a revision.

A: Thank you for this very positive comment.

Figure 1: the authors should ensure the representation of every measurement point with its corresponding error bar. This would provide a more comprehensive and accurate visualization of the data, facilitating a clearer understanding of the precision and variability associated with each measurement.

A: We appreciate this constructive suggestion and feel sorry for this inconvenience. As suggested, we now revised our Fig. 1. In the revised figure, the symbols representing all the data points have been changed to very small ones. In this

way, some of the very small error bars can be seen. Additionally, we have included the source data used for calculating all the curves.

The authors mentioned in the manuscript that, "Although the tigecycline molecule (Tig1) on the 39S mitoribosome was not at high resolution, it was still sufficiently resolved for an unambiguous assignment (Fig. 2b)." Despite the lack of explicit evidence in Figure 2b to support the resolution-based assignment, the observed binding site on the 55S mitoribosome strongly suggests that this density corresponds to a tigecycline molecule. The authors should improve the segmentation of the map in this region and provide a clear model-to-map fitting representation. Additionally, it might be beneficial for the authors to include a discussion sentence acknowledging that the assignment of this density on the 39S mitoribosome was based on the observed binding site on the 55S. Suggestions also extend to improving the overall clarity of Figure 2.

A: Thank you for this very constructive suggestion. As suggested, we have enhanced the segmentation of the map in this tigecycline region in our revised manuscript (new Supplementary Figure 5). Additionally, we also have included the segmentation of maps in this tigecycline region from the 39S mitoribosome treated with 5 μ M and 10 μ M tigecycline.

Furthermore, we have modified our description as follows:

"Unfortunately, the resolution of the tigecycline molecule (Tig1) on the 39S mitoribosome was not optimal. However, based on the assignment of the corresponding Tig1 molecule on the 55S mitoribosome, we attempted to identify this additional density in the same region as a tigecycline molecule in the 39S mitoribosome (Supplementary Fig. 5b)."

Concurrently, we have made revisions to Fig. 2 to improve the overall clarity.

Jenner et al., 2013 (reference 11 in this manuscript) determined the X-ray crystallography Structure of the 70S-tigecycline complex, which revealed the primary binding site. The authors identified four distinct binding sites in this study - despite the potential conservation between bacterial and human mitochondrial ribosomes. Can they further comment on this discrepancy?

A: We appreciate this suggestion. In general, the conservation between bacterial and human mitochondrial ribosomes is actually limited. Although the mitochondrial ribosome is closer related to the bacterial one than the human 80S ribosome, it has considerably diverged as illustrated by its much smaller rRNA content, distinct protein composition and overall different appearance. Differential interaction patterns with small molecules are thus not very surprising. Therefore, in addition to the discussion we had on Tig1, we have incorporated more discussion in our revised manuscript regarding the observed discrepancies between our 55S-tigecycline complex and the 70S-tigecycline complex, specifically pertaining to Tig2-4.

"Undoubtedly, the Tig1 binding site on the 55S/39S mitoribosome is an interesting site. It has never been observed before, including on the bacterial 70S ribosome and the eukaryotic 80S ribosomes. A previous extensive biochemical study on TcmX has already demonstrated that the non-canonical U-U base pair in the PTC determines the binding property of TcmX²². It is also plausible to speculate that the non-canonical C-C base pair may be able to determine the binding property of tigecycline. Based on available rRNA data, in the bacterial world, 63% of species have a corresponding C-C base pair at this position, whereas only 37% have a U-U base pair required for TcmX binding²². This could explain the much broader antibacterial spectrum of tigecycline compared to TcmX. Therefore, a combination treatment with both tigecycline and tetracenomycin X may serve as a new therapy with a much broader effect.

While the Tig3 molecule binds to the conserved "primary" binding site as in the bacterial 70S ribosome, the Tig2 molecule is exclusively observed binding to H70 of the 16S rRNA in the human 55S mitoribosome. This distinction may arise from the lack of conservation of H70 between the 55S mitoribosome and the bacterial 70S ribosome, particularly the significant A934 stacking with the tigecycline rings (C1941 in *E. coli*). The Tig4 molecule on the other hand is only detected in the human 55S mitoribosome treated with 100 μ M tigecycline, suggesting that Tig4 binds only under high concentrations of tigecycline."

In the X-ray structure, the ratio between tigecycline and ribosome was at a 20-fold excess, while here, it is unclear what ratio was used for the cryo-EM data. The authors may consider disclosing the ratio used in their experimental conditions.

A: We apologize for the lack of clarity. As suggested, we have included the ratio used in our experimental conditions (new section: Cryo-EM grids preparation) in the revised methods section.

"Factor-free/empty 80S ribosomes (140 nM) were incubated with either 4 μ M or 100 μ M tigecycline (~28.6- or 714.3-fold excess) for 45 min on ice before plunge freezing. 3.5 μ L of sample was applied to precoated (3 nm carbon) R3/3

carbon support grids (Quantifoil) with 45 s pre-blotting and 3 s blotting time. Native human ribosomes (OD260 \approx 5, approximately 0.1 μ M) were incubated with 100 μ M tigecycline (\sim 1000-fold excess) for 15 min on ice. Human mitoribosomes (OD260 \approx 5, approximately 0.16 μ M) were incubated with 5 μ M, 10 μ M, and 100 μ M tigecycline (approximately 31.25, 62.5, and 625-fold excess relative to the mitoribosome, respectively) for 15 minutes on ice. The latter samples (3.5 μ L) were applied to precoated (2 nm) R1.2/1.3 carbon-supported copper grids (Quantifoil), blotted for 4 s at 4 $^{\circ}$ C, and plunge-frozen in liquid ethane using an FEI Vitrobot Mark IV.”

One should also inquire whether the authors would detect these secondary binding sites at concentrations of tigecycline that are clinically relevant. Addressing this point would provide valuable insights into the practical implications of the identified binding sites and their potential relevance in the toxicity of tigecycline.

A: We appreciate this constructive suggestion. The concentration of 100 μ M was selected based on considerations of achieving robust structural data and capturing the binding interactions effectively. However, we recognize the importance of conducting experiments at physiologically relevant concentrations to enhance the clinical significance of our findings.

In response to this concern, we made an effort to perform additional experiments using lower tigecycline concentrations that align more closely with physiological conditions. Although there is not a very consistent physiological concentration of tigecycline in our body after the treatment, numerous studies (see references below) suggest an average serum concentration ranging from 4 to 10 μ M. Notably, certain tissues exhibit significantly higher concentrations, such as an 8.6-fold increase in the lung, a 38-fold increase in the gall bladder, a 78-fold increase in alveolar cells, where tigecycline concentrations can accumulate to several hundred μ M.

As a response, we have performed two additional structural analysis of the mitoribosome at tigecycline concentrations of 5 μ M and 10 μ M. In these new structures, Tig1, Tig2, and Tig3 are clearly observed in the 55S mitoribosomes treated with both 5 μ M and 10 μ M tigecycline. Unfortunately, the presence of Tig4 is not as distinct in these new structures, suggesting its functional relevance primarily under high tigecycline concentrations. We also specify this in the following description of Tig4 molecule. We have incorporated these findings into our revised manuscript (Figure 2, Results sections 2, 4, Discussion and supplementary figure 5) and discussed the observed differences as followed:

Results sections 2:

“In contrast, when the human mitoribosomes were exposed to 5 μ M or 10 μ M tigecycline, concentrations within the physiological range during clinical usage 23-25, only three tigecycline molecules (Tig1-3) could be distinctly assigned, with Tig4 absent (Fig. 2a, b; Supplementary Fig. 5a). This discrepancy suggests that the binding of Tig4 occurs specifically under high concentrations of tigecycline.”

Results sections 4:

“Tig4 binds to a new region near h1 and h44 of the 12S rRNA (Fig. 4d), exclusively observed in the 55S mitoribosome treated with 100 μ M tigecycline.”

Discussion:

“The Tig4 molecule on the other hand is only detected in the human 55S mitoribosome treated with 100 μ M tigecycline, suggesting that Tig4 binds specifically under high concentrations of tigecycline.”

In addition, we also performed cryo-EM single particle analysis with empty human 80S ribosomes at concentrations of 4 and 100 μ M tigecycline. We find that binding to 80S ribosomes occurs at high concentration but not at all at the clinically relevant concentration. Thereby showing clear concentration dependent binding differences between cytoplasmic and mitochondrial ribosomes. This is in agreement with our biochemical data (Fig 1) and highlights again that the toxicity of tigecycline at a clinically relevant concentration is primarily due to mitoribosome targeting. Accordingly, we added a new paragraph to the results section (“**Tigecycline does not bind human 80S ribosomes at physiological conditions**”) and a revised Figure 5.

These additional structural analyses validate our observations under conditions that better emulate the therapeutic environment, providing a more comprehensive understanding of tigecycline binding and its physiological implications.

Reference:

1. https://www.accessdata.fda.gov/drugsatfda_docs/nda/2005/21-821_Tygacil.cfm
2. Rodvold, K. A. et al. Serum, tissue and body fluid concentrations of tigecycline after a single 100 mg dose. *J Antimicrob Chemother* 58, 1221-1229, doi:10.1093/jac/dk1403 (2006).

3. Meagher, A. K., Ambrose, P. G., Grasela, T. H. & Ellis-Grosse, E. J. The pharmacokinetic and pharmacodynamic profile of tigecycline. *Clin Infect Dis*, 41, Suppl 5, S333-340, doi:10.1086/431674 (2005).
4. Brink, A. J. et al. Guideline: appropriate use of tigecycline. *S Afr Med J* 100, 388-394, doi:10.7196/samj.4109 (2010).

Figure 1 presents the concentration-dependent inhibition of the 80S ribosome by tigecycline. Although the inhibition appears mild, the structure reveals the presence of six molecules of tigecycline bound to the 80S ribosome. Can this discrepancy be explained by the concentration of the antibiotic used for cryo-EM? The authors used a 1000-fold excess of tigecycline compared to the ribosome (approximately 0.1 μM vs. 100 μM). This point needs further discussion in the manuscript. The manuscript presently lacks clarity regarding the specificity of these interactions.

A: Yes, this discrepancy can be explained by the different concentrations used. As pointed out by the reviewer our biochemical data suggests only a very mild inhibition of human 80S by tigecycline. Consequently, when preparing the native 80S sample a higher concentration (170 μM) was used to treat cells in order to achieve translational inhibition relevant to 80S ribosomes, while at the same time ensuring structural robustness. Secondly, this high concentration is equivalent to the concentrations used to treat yeast cells for our previously published yeast 80S + tigecycline data set that we use here for comparison. Furthermore, the same concentration has also been used for ribosome profiling in yeast before (references #33 and #34 in the manuscript). Lastly, while such high doses might not be clinically relevant, they already have been used to enrich and stabilize translation intermediates and may be of further use in future research. Therefore, the structural characterization of the generated intermediates appears as valuable information for the scientific community.

Meanwhile, to address the issue raised, as already stated above, we analyzed two new cryo-EM datasets of factor-free/empty human 80S incubated with 4 μM and 100 μM tigecycline, equivalent to \sim 28.6 or 714.3 times excess, respectively. We observed no tigecycline molecules at low concentration, only at the higher concentration which also clears the discrepancy between our biochemical and structural data. Accordingly, we added a new chapter to the results section (“**Tigecycline does not bind human 80S ribosomes at physiological conditions**”) and revised Figure 5. To improve clarity, the excess over ribosomes is clearly stated in the Methods section (see section: **Cryo-EM grids preparation**) and the concentration used for the native human and yeast 80S is now also stated in the main text:

“To further explore the effect of tigecycline on cytoplasmic translation, we purified native 80S ribosome complexes from human HEK293 cells treated with a high dose of tigecycline (170 μM).”

and

“To validate these findings on the human 80S ribosome, we also solved the cryo-EM structure of the tigecycline complex with the yeast 80S ribosome from one of our published datasets³³ (Fig. 6; Supplementary Fig. 11; Supplementary Tables 2-3), where a similar high-dose treatment (170 μM) was used.”

We also clearly state that these secondary binding site could be the result of the incubation with high concentration of tigecycline in the discussion as follows:

“The existence of these “secondary” binding sites most possibly being a result of incubation with high concentrations of tigecycline (100 μM), however, our observations reveal conserved binding sites between human and yeast. Despite the non-physiological concentration of tigecycline used, studying the inhibition mechanism at high concentrations remains valuable for research purposes³⁴ or to guide new drug design.”

Reviewer #1 (Remarks to the Author):

Following the reviewers' advice, the authors have significantly improved the work. I have no additional suggestions.

Reviewer #2 (Remarks to the Author):

The authors have adequately addressed all my concerns regarding the manuscript. Consequently, I have no objections to its publication in Nature Communications.